# Single molecule high-throughput footprinting of small and large DNA ligands

Maria Manosas[1,2], Joan Camunas-Soler[1,2,4], Vincent Croquette[3] & Felix Ritort[1,2]

Most DNA processes are governed by molecular interactions that take place in a sequence-specific manner. Determining the sequence selectivity of DNA ligands is still a challenge, particularly for small drugs where labeling or sequencing methods do not perform well. Here, we present a fast and accurate method based on parallelized single molecule magnetic tweezers to detect the sequence selectivity and characterize the thermodynamics and kinetics of binding in a single assay. Mechanical manipulation of DNA hairpins with an engineered sequence is used to detect ligand binding as blocking events during DNA unzipping, allowing determination of ligand selectivity both for small drugs and large proteins with nearly base-pair resolution in an unbiased fashion. The assay allows investigation of subtle details such as the effect of flanking sequences or binding cooperativity. Unzipping assays on hairpin substrates with an optimized flat free energy landscape containing all binding motifs allows determination of the ligand mechanical footprint, recognition site, and binding orientation.

[1] Small Biosystems Lab, Departament de Fsica de la Matèria Condensada, Facultat de Física, Universitat de Barcelona, Barcelona 08028, Spain. [2] CIBER de Bioingeniería, Biomateriales y Nanomedicina, Instituto de Salud Carlos III, Madrid 28029, Spain. [3] ENS, PSL Research University, UPMC, Université Paris Diderot, Dept. de Physique et IBENS, CNRS UMR-8550, LPS, 24 rue Lhomond, 75231 Paris, France. [4] Present address: Departments of Bioengineering and Applied Physics, Stanford University, Stanford, CA 94304, USA. Maria Manosas and Joan Camunas-Soler contributed equally to this work. Correspondence and requests for materials should be addressed to F.R. (email: fritort@gmail.com)

Many molecules bind to DNA in a sequence-specific manner. The most ubiquitous example are proteins involved in processing DNA and gene regulation, which can recognize DNA sequences of up to 15 base-pairs (bp) in order to carry out their function (e.g., unwind, overwind, copy, cut and paste DNA). Besides, many natural and synthetic drugs used in chemotherapies also bind DNA, changing its structure and interfering with such processes. These drugs are mostly small ligands that target short DNA sequences (typically 2–5 bp) via minor or major groove binding, intercalation and/or alkylating activity[1–3]. Hence, determining the selectivity and affinity of ligands (both small drugs and large proteins) to different DNA sequences is key to understand fundamental biological processes and for its potential applications in drug development[4, 5].

The classic approach to determine ligand binding sites to DNA are footprinting experiments[6, 7]. In DNA footprinting the binding sequence and coverage size of a ligand are determined from the restriction pattern of a radiolabelled DNA molecule that has been incubated with the ligand, achieving up to one base-pair (bp) resolution[8, 9]. Although affinities and kinetic rates can also be inferred, quantitative techniques such as surface plasmon resonance (SPR), isothermal titration calorimetry (ITC), UV-melting, electrophoretic mobility shift assays (EMSA), or thermoforesis are better suited to perform thermodynamic and kinetic measurements[10–12]. In the last years the development of high-throughput techniques such as protein binding microarrays (PBM), HT-Selex, or MITOMI has allowed to screen the selectivity of proteins against a large numbers of DNA sequences using oligonucleotides to measure relative affinities and binding kinetics[4, 13–15]. However, current high-throughput techniques are typically based on either DNA sequencing or fluorescence

labeling for detection, making their use particularly challenging to characterize the binding of small ligands and drugs[14, 16].

In this work, we propose a single-molecule footprinting assay that allows detecting the binding sites of a ligand (drug or protein) with a resolution of ~2 bp, faster and with less reactants consumption than with most bulk techniques. Moreover, a single assay can be used to determine both the thermodynamics and kinetics of binding, which usually requires multiple experiments and techniques. The method does not require the use of tags or labeling on the ligand, and is therefore particularly suited for small ligands. The sensitivity of the assay allows investigating subtle details such as the presence of multiple and cooperative ligands binding to DNA or the effect of ligand orientation in the mechanical stability of the complex.

Our method is inspired in previous work by the group of M. Wang[17–20] and ourselves[21–23] and is based on sensing ligand binding by means of a DNA unzipping assay[24–26]. Magnetic tweezers[26–28] are used to apply a mechanical force at the opposite ends of a DNA hairpin disrupting the base-pairing interactions and unraveling the double helix structure[29, 30]. By applying forces above ~15 pN, the DNA hairpin is unzipped cooperatively in a single step. Addition of ligands leads to a multi-step unzipping with several blockages generated by ligand binding, allowing to determine the ligand sequence selectivity. Binding kinetics and thermodynamics can be studied by measuring binding lifetimes at different forces. To perform these measurements in an unbiased way, we propose a way of generating sequence-optimized hairpins with a flat free energy landscape that allow us to quantitatively test all possible tetramers (256 combinations) in a hairpin 200 bp long. By using these engineered DNA hairpins (~0.1–10 kbp) a large number of potential binding sequences can be tested in a single

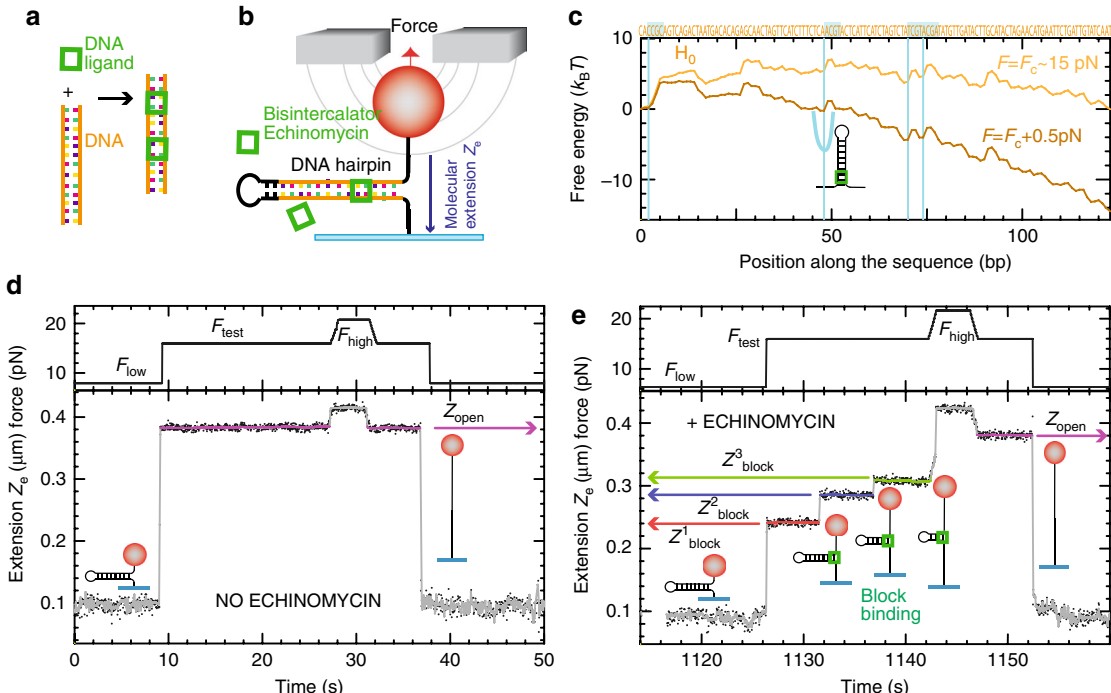

**Fig. 1** High-throughput footprinting assay using magnetic tweezers. **a** Schematics of the ligand binding to DNA. **b** Schematics of the magnetic tweezers experimental set-up. **c** Free energy landscape of hairpin $H_0$ at the coexistence force ($F_c = 15$ pN) in *yellow* and at a larger force ($F_c + 0.5$ pN) in *brown*. The $H_0$ sequence is shown in the upper part and the tetranucleotide XCGY motifs, that are the preferred Echinomycin binding sites, are highlighted in blue. **d** Experimental trace $Z_e(t)$ obtained with $H_0$ when applying a FC protocol, showing the unzipping and rezipping of the hairpin (*lower panel*). The FC protocol consists in the alternation of three different forces: $F_{low}$, $F_{test}$, and $F_{high}$ (*upper panel*). **e** Experimental trace $Z_e(t)$ obtained with $H_0$ in presence of [Echinomycin] = 300 nM (*lower panel*) when applying a FC protocol (*upper panel*), showing the three different binding events during the hairpin unzipping ($Z_{block}^{1,2,3}$, *red, blue, green* respectively). The extension of the fully unfolded state at the same force $F_{test}$ is also measured ($Z_{open}$, *purple*)

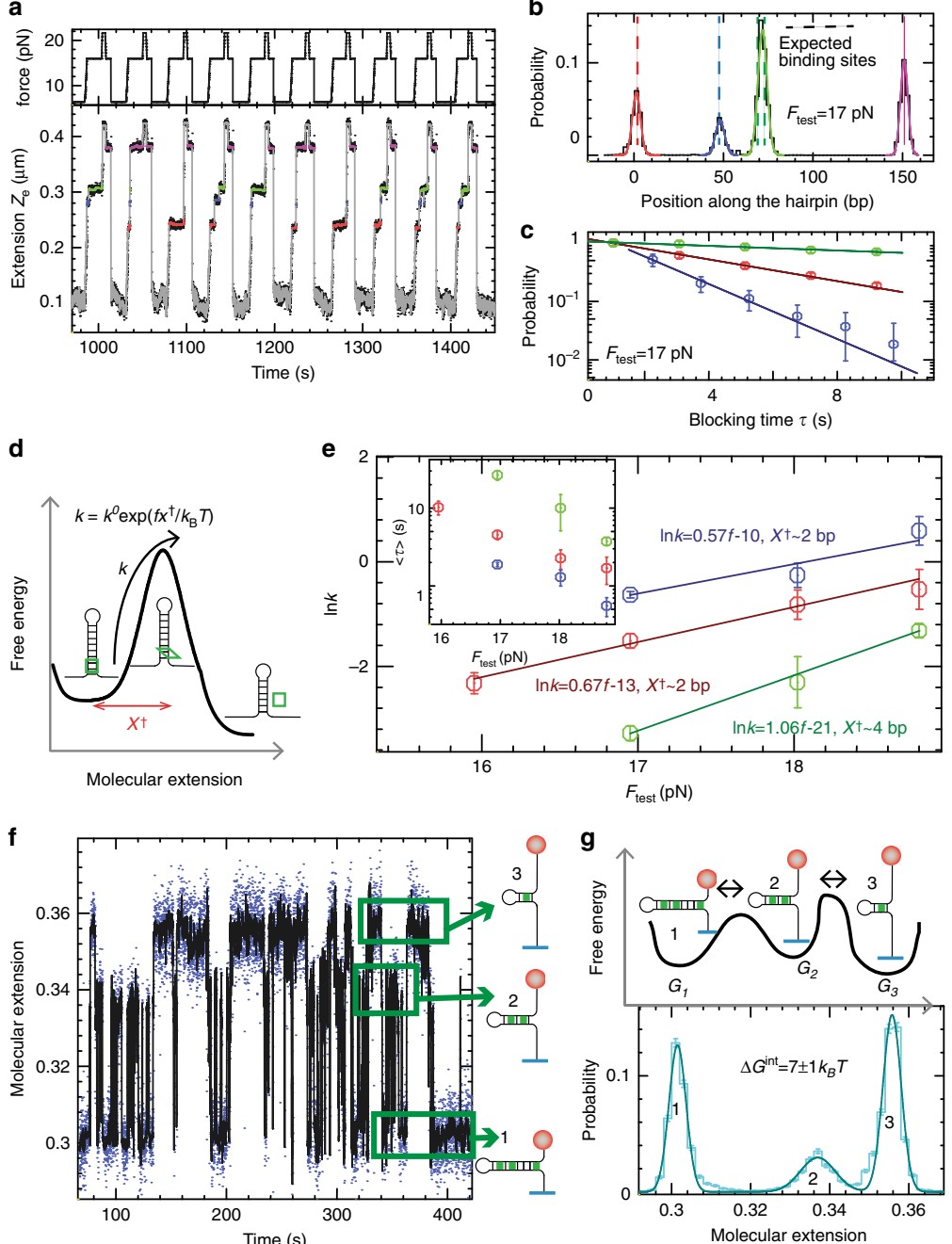

**Fig. 2** Binding selectivity, kinetics, and thermodynamics of Echinomycin. **a** The FC protocol in presence of Echinomycin at 300 nM is repeatedly applied to unzip $H_O$ DNA hairpin and detect blocking events, shown in *red*, *blue*, and *green*. **b** Histogram of molecular extension in phase 2 converted to base-pairs, for the bead in **a**, shows three peaks. The histogram is fitted to Gaussian functions centered around the three binding sites. *Results* shown correspond to 135 cycles. **c** Distribution of blocking times, $\tau$, for the three binding sites at $F_{test} = 17$ pN for the bead in **a**. *Error bars* are inversely proportional to the square root of the number of points for each bin. **d** Schematics of ligand unbinding described as a Kramer Bell-Evans activated process. **e** Logarithm of the unbinding rate, $k = 1/\langle\tau\rangle$ as a function of the applied force, $F_{test}$, for the three binding sites. The $k$ and $\langle\tau\rangle$ values are computed as the average over 10 beads. *Error bars* are the s.e.m. The result of the linear fits are also shown. (Inset) Average blocking time, $\langle\tau\rangle$, as a function of $F_{test}$ for the three binding sites. **f** Equilibrium extension trace for $H_O$ in the presence of Echinomycin at force $F_{hop} = 15.5$ pN, showing hopping between the three partially unzipped configurations blocked at the Echinomycin binding sites. The three configurations are represented at the right side of the panel. **g** Schematics of the free energy landscape of the hairpin in presence of Echinomycin that shows three minima corresponding to the three partially unzipped hairpin configurations (*upper panel*) and histogram of the molecular extension shown in **f**, presenting three peaks, that is fitted to Gaussian functions (*lower panel*). The relative weights of the Gaussian are used to compute the probabilities of the different configurations and to estimate the Echinomycin binding energy (Methods and Supplementary Table 3)

assay in which the ligand "mechanical" footprint, recognition site and binding orientation can be determined in a single experiment. Moreover, since magnetic tweezers present a large level of parallelization (simultaneous tracking of ~100

experiments, Supplementary Fig. 1)[26, 31–34], the method allows us to obtain large statistics in short times (~hour). In this way, single-molecule footprinting expands the current repertoire of DNA binding assays based on the mechanical unfolding of DNA

hairpins[17–20, 24–26, 35–37], being particularly useful to quantitatively study the binding of small ligands to DNA.

## Results

**Experimental set-up and DNA hairpin design.** The experimental configuration consists of a DNA hairpin tethered between a glass surface and a micron-sized paramagnetic bead (Fig. 1a, b). A pair of permanent magnets generates a strong magnetic field gradient that pulls the bead, and so the tethered DNA molecule, with pN forces (Methods section). Detection of the 3D bead position allows determining the extension of the tethered molecule and the applied force[28]. The hairpin unzips above a threshold force, typically ranging between 13 and 16 pN (depending on DNA sequence and ionic conditions), leading to the release of single-stranded DNA (ssDNA), which increases the length of the DNA tether by ~1 nm for each bp unwound at ~15 pN. Consequently, by recording the time evolution of the DNA extension, $Z_e(t)$, the unzipping reaction can be followed in real time. Since ligands that bind to DNA locally stabilize the double helix structure (typical binding energies are in the range of ~5–15 $k_BT$), their binding along the DNA hairpin can be detected as kinetic roadblocks observed during the unzipping reaction. Key for the method is the fact that, in the absence of ligands, the hairpin unzips in a single step without intermediates. For this, the DNA hairpin is engineered to have a uniform stability along the sequence, presenting a free energy landscape, defined as the free energy as a function of the number of bps unzipped, as smooth as possible with low barriers (Fig. 1c and Methods section). The uniform DNA stability ensures that the observed unzipping roadblocks exclusively correspond to ligand binding events and not to sequence-dependent effects, and that the mechanical applied strain affects the different potential binding sites along the DNA sequence in a homogeneous way.

**Echinomycin as a paradigmatic case.** Proof of principle selectivity assays have been performed with a ~150 bp DNA hairpin (Methods and Supplementary Fig. 2), referred as $H_0$, and the bis-intercalator Echinomycin, an anti-tumoral drug with specificity for CpG steps[38, 39]. In absence of Echinomycin, hairpin $H_0$ unzips cooperatively when applying a force $F_{test}$ above the coexistence force $F_c$ (force at which the free energies of the folded and fully unzipped hairpin are equal), Fig. 1d. By decreasing the force to $F_{low} < F_c$, the hairpin refolds. When adding Echinomycin, the unzipping of hairpin $H_0$ occurs in multiple steps presenting blockages at intermediate positions before reaching the maximal extension $Z_{open}$ (extension of the fully unzipped hairpin), Fig. 1e.

In order to perform systematic DNA ligand selectivity measurements, we developed a force-cycle (FC) protocol in which three force values ($F_{low}$, $F_{test}$, $F_{high}$) are alternated in 5 different phases (Figs. 1e and 2a): (1) the force is first settled to a low value ($F_{low}$ ~ 6 pN) where the hairpin is stable and the ligand can bind to the duplex DNA; (2) the force is increased up to $F_{test}$ ~ 17 pN to detect blockages at the position where the ligand is bound (three blockages -red, blue and green- are observed in Fig. 1e); (3) the force is increased a higher force value, $F_{high}$ ~ 25 pN, to remove any remaining ligands bound to the template and reach the fully unfolded configuration; (4) the force is reset to $F_{test}$ for a few seconds to obtain the reference extension ($Z_{open}$) of the fully unfolded hairpin; (5) the force is decreased to $F_{low}$ to induce the hairpin rezipping and start a new cycle. For all ligands studied in this paper, the force $F_{high}$ ~ 25 pN is enough to force-unbind the ligand from the hairpin in most cycles. This fact is supported by optical tweezers studies of the kinetics of some of these ligands binding to short DNA hairpins[23]. For ligands showing stronger affinities to the hairpin, the protocol can be modified by applying

either (i) a longer $F_{high}$ step, or (ii) a higher force (up to 100 pN using 2.8 µm beads). A similar approach can be used to increase binding lifetime statistics in the $F_{test}$ step.

The DNA sequence where the ligand binds can be determined from the measured extension $Z_e$, performing differential measurements, to reduce drift (Methods and Supplementary Fig. 3) and using a conversion factor from measured distances (in nanometers) to positions (in bps) (Supplementary Fig. 4). From cycle to cycle, the blockages are observed at the same locations, showing that Echinomycin is binding to specific positions along the DNA sequence (Fig. 2a). By repeating the FC protocol several times (>100), for ~50 beads in parallel, we collected statistics of the blockages. For each bead, the histogram of molecular extension in phase 2 ($F_{test}$), shows peaks very close to the position of the CpG steps along the sequence (Fig. 2b), in agreement with previous results[38, 39]. The histogram can fitted to Gaussian functions to determine the binding positions with ~2 bps resolution (the width of the Gaussian, Supplementary Table 2).

We also measure the blocking time $\tau$ for each binding event, which mean value is directly related to the local stability and kinetic barrier for ligand unbinding. The blocking time probability distribution follows a single exponential for the three binding sites (red, blue, and green, Fig. 2c), indicating a single rate-limiting step for Echinomycin unbinding under unzipping force. The unbinding reaction can be described as a Kramer Bell-Evans activated process, where the kinetic rate $k$ for unbinding follows: $k = 1/<\tau> = k_0 \exp(fX^{\dagger}/k_BT)$ (Methods and Fig. 2d), with $X^{\dagger}$ being the distance from the transition state to the ligand bound state and $k_0$ being the extrapolated ligand unbinding rate at zero force. The larger average lifetime of the third blockage (Fig. 2e, inset) can be explained by the presence of two contiguous XCGY sites at this region, suggesting that two bis-intercalators are consecutively bound increasing the local stability of the site.

The dependence of the average block lifetime on the applied force is measured by repeating the FC protocol at different values of $F_{test}$. As expected for a Kramer-like activated process, the logarithm of the unbinding rate $k$ follows a linear behavior with $F_{test}$ (Methods and Fig. 2e). Using $X^{\dagger}$ extracted from the slope of the linear fits, we estimate that the transition state is located about 2 nm from the bound state for the first and second binding sites, corresponding to the extension of two unwound bps. If the bis-intercalator intercalates the two rings between the CpG step stabilizing 4 bps, XCGY, the transition state would correspond to a half-intercalated state. In contrast, the extracted $X^{\dagger}$ for the last binding site, where presumably two contiguous bis-intercalators are bound, is 4 nm, or 4 unwound bps, which corresponds to the extension change for the unbinding of a single bis-intercalator, suggesting that the binding of two contiguous Echinomycin bis-intercalators is cooperative in agreement with recent results[23]. These values for $X^{\dagger}$ for echinomycin are significantly larger than those previously obtained for restriction endonucleases in dynamic force-spectroscopy experiments ($X^{\dagger} \leq 1$ nm)[18], showing that small ligand interactions are more sensitive to force-induced unbinding than large protein-DNA complexes. The unbinding rate extrapolated at zero force, $k_0$, extracted from the fits lies in the range of $4 \times 10^{-5} – 2 \times 10^{-6}$ s$^{-1}$ for a single bis-intercalator, a value compatible with previous measurements of neutral bis-intercalators[21, 40, 41].

When exploring the unbinding kinetics of Echinomycin at different stretching forces, we found a small force range of applied forces where the DNA molecule spontaneously hops between three configurations (referred as 1,2,3), each one corresponding to the partially unzipped hairpin blocked at one

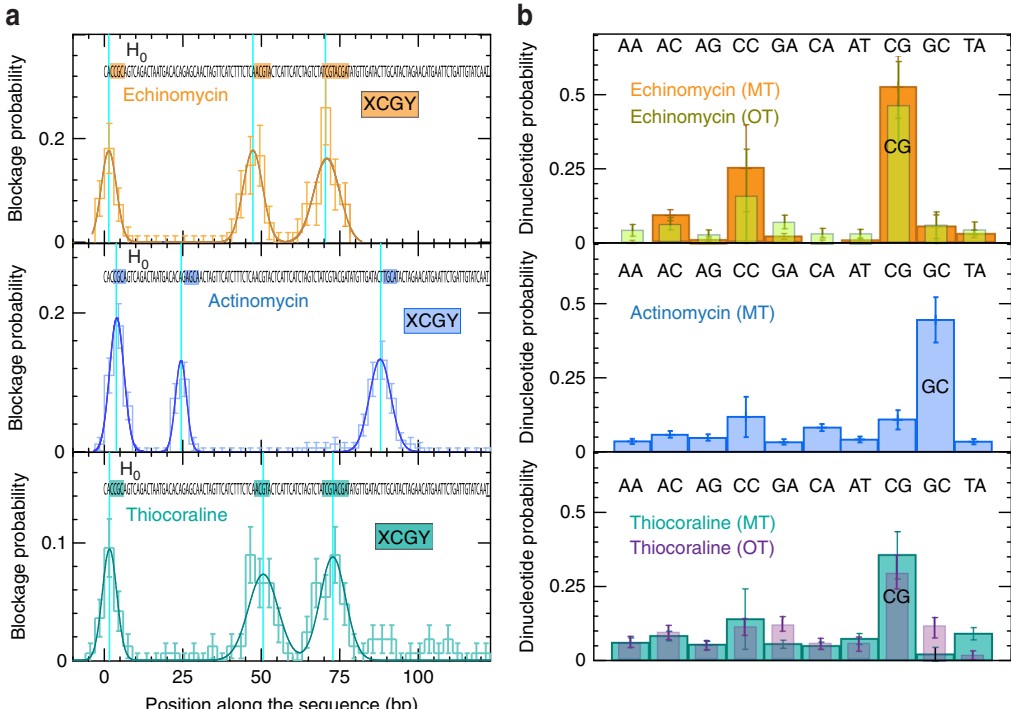

**Fig. 3** Binding selectivity measurements for different intercalators. **a** Distribution of blockage positions obtained by applying the FC protocol on $H_0$ in presence of Echinomycin at 300 nM (*upper*, number of beads = 93), Actinomycin at 1 µM (*middle*, number of beads = 174), and Thiocoraline at 100 nM (*lower*, number of beads = 127). The blockage position for each bead is obtained from 50 to 300 cycles of the FC protocol. *Error bars* are inversely proportional to the square root of the number of points for each bin. The distribution is fitted to Gaussian functions. **b** The probability of observing a blockage at each possible dinucleotide step for Echinomycin (*upper*), Actinomycin (*middle*), and Thiocoraline (*lower*), computed from data in **a**. The results for Echinomycin and Thiocoraline obtained with optical tweezers are also shown for comparison (Supplementary Fig. 6 and ref. [21]). *Error bars* are inversely proportional to the square root of the number of points for each bin

of the three binding sites (Fig. 2f). From these "equilibrium" hopping traces, at fixed force $F_{hop}$, we can compute the probability of populating each partially unzipped configuration (Fig. 2g), which is related to their free energy differences at $F_{hop}$ (Methods). By evaluating the different free energy contributions of the three partially unzipped configurations, as described in Methods section, we can extract an average binding energy of Echinomycin to the XCGY sites of ~7 $k_B T$ (Supplementary Table 3).

**Footprinting of small ligands recognizing different motifs.** In order to validate the approach proposed here, we use the FC protocol to determine the selectivity of three different small ligands: Echinomycin, Actinomycin, and Thiocoraline. Previous studies have shown that bis-intercalators Echinomycin and Thiocoraline have a preference to bind CpG steps[21, 38, 42], whereas the mono-intercalator Actinomycin has a preference to bind GpC steps[43]. For each intercalator, we perform the FC protocol for several beads in parallel. For each bead, we compute the blockage positions, estimated as the center of the peaks in the histogram of molecular extension during phase 2 (as done for Echinomycin in Fig. 2b). We next build the distribution of blockage positions obtained for all beads (~100–200 beads), that show a series of peaks centered at specific DNA locations (Fig. 3a and Supplementary Fig. 5). To determine the preferred binding sequences from our experiments, we assign to each blockage the nearest bp position along the DNA sequence and compute the probability of observing a blockage at each possible dinucleotide step (Fig. 3b), finding the sites previously determined with other techniques[21, 38, 42, 43]. Optical tweezers can be also used to perform similar unzipping experiments[17, 21],

recovering analogous results (Fig. 3b and Supplementary Fig. 6). Interestingly, for Echinomycin, we observe that the two first peaks have a width of ~2 bps whereas the third peak, where two consecutive sites are present, is larger (~4 bps) (Supplementary Table 2). A similar effect is observed for Thiocoraline, that shows blockages events along all the sequence (Fig. 3a), and where the three main binding peaks (sequence-specific binding to CG steps) have a larger width than expected (~3–4 bps, Supplementary Table 2). These results seem compatible with the known lower specificity of Thiocoraline in relation to Echinomycin[21, 42], and suggests that multiple binding sites at nearby positions might be inferred from the width of the mechanical footprints in an analogous way to what is observed in bulk footprinting experiments.

**Design of optimal DNA hairpins for multiple site testing.** Hairpin $H_0$ was specially designed to have a flat free energy landscape and to test a set of well-characterized ligands, so that their preferred binding sites appeared several times along the sequence. However, in order to test a ligand with unknown recognition sequence, a hairpin containing the ligand recognition sequence plus all possible competing sequences should be used. Moreover, in order to get unbiased results, we want to avoid large differences in the frequency the different sites appear along the sequence (Methods section). Consequently, to test the binding selectivity of any ligand that binds or recognizes 4 or less bps, such as many enzymes and drugs do, we aim at generating sequences that have all combination of 4 bases at least once and not more than twice (Fig. 4a). Sequences verifying this condition and presenting a flat free energy landscape have been generated by a Monte Carlo simulation by minimizing the height

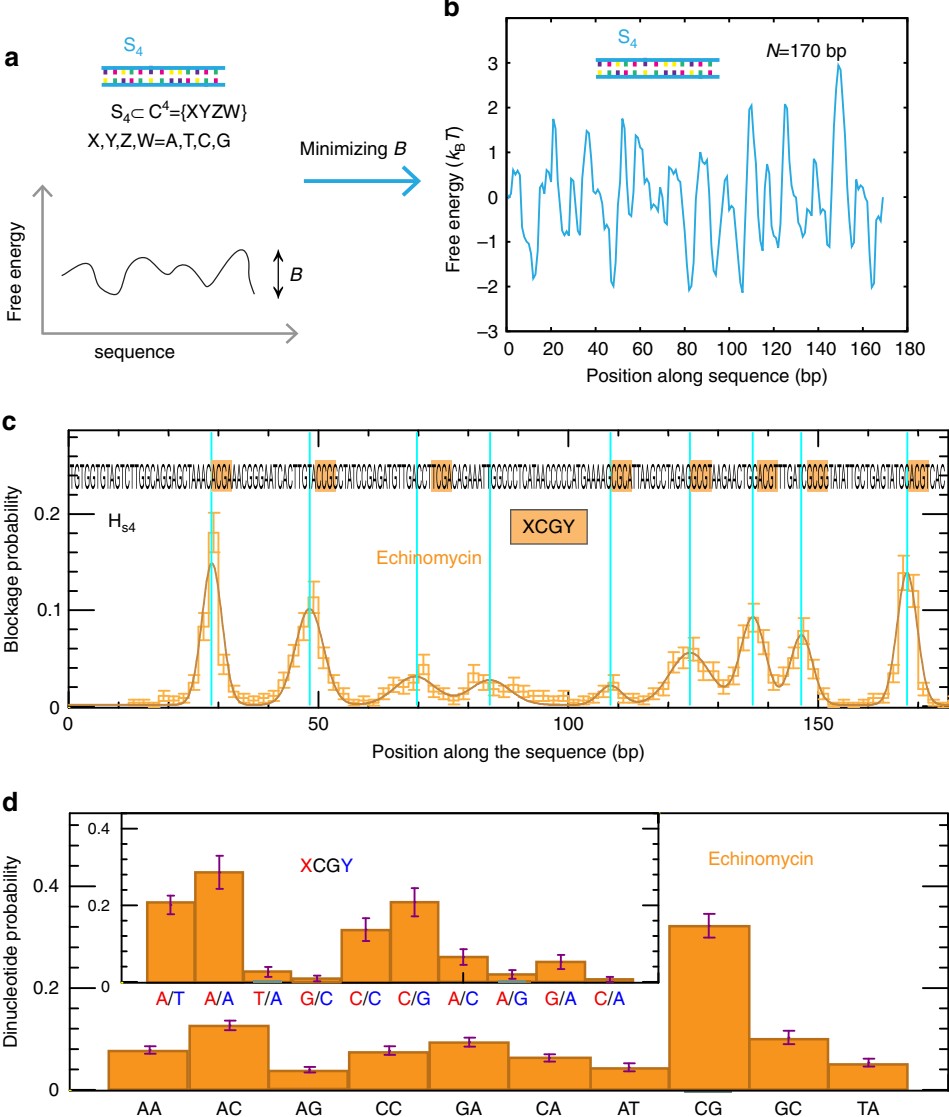

**Fig. 4** Selectivity measurements in optimized DNA sequences. **a** Schematics of the design of an optimized sequence $S_4$ for selectivity measurements of ligands that bind or recognize 4 bps or less. $S_4$ includes all tetranucleotide combinations ($K = 4$). **b** Free energy landscape of $S_4$ sequence, containing 170 bps, obtained by Monte Carlo minimization of the roughness of the unzipping free energy landscape (Methods section). **c** Distribution of blockage positions obtained applying the FC protocol with hairpin $H_{s4}$ in presence of Echinomycin (number of beads = 182). Error bars are inversely proportional to the square root of the number of points for each bin. **d** The probability of observing a blockage at each possible dinucleotide step obtained from the results presented in **c**. The *inset* shows the probability of observing a blocking event at XCGY positions with X, Y = A,T,C,G. *Error bars* are inversely proportional to the square root of the number of points for each bin

of the barriers in the unzipping free energy profile (Fig. 4b, Supplementary Fig. 7 and Methods section).

Echinomycin as a bis-intercalator clamps the two aromatic rings between two consecutive bps and presents a preference for the GpC step. However, neighboring flanking bps play also a role in the binding stability, since they have base-stacking interactions with the intercalated rings. A hairpin, named $H_{s4}$ (Methods section), that includes one of the aforementioned optimized sequences, $S_4$, has been used to test the the effect of flanking bps on Echinomycin-DNA stability. The FC protocol with $H_{s4}$ in presence of Echinomycin leads to unzipping blockages (Supplementary Fig. 8), that correlate to the location of CpG steps (Fig. 4c, d). The probability of blockages in XCGY position, as a function of the XY flanking bps (X, Y = A,T,C,G,C), shows a preference for AT, AA,CC, and CG flanking bps (inset Fig. 4d). In these experiments, the presence of wider peaks seem to correlate with the presence of CC or GG stretches

(Supplementary Table 2), that are potential secondary sites for Echinomycin[38], [39]. This might be particularly important in regions containing a GC-rich stretch ahead of a strong CG binding motif (e.g., CCTTCGA, peak 3) where two echinomycin ligands might be bound one next to each other (i.e., to the dinucleotides CC and to CG), as observed in previous bulk footprinting experiments[38].

**Mechanical footprinting of enzymes shows binding orientation.** For small ligands, such as intercalators, the unzipping blockage positions are directly related to its binding or recognition sequence. However, for large ligands, such as sequence-specific DNA-binding proteins, the recognition sequence is typically shorter (2–5 bps) than the total extension of DNA covered by the ligand (10–30 bps). For the latter, the unzipping blockage positions will be related to the ligand/DNA

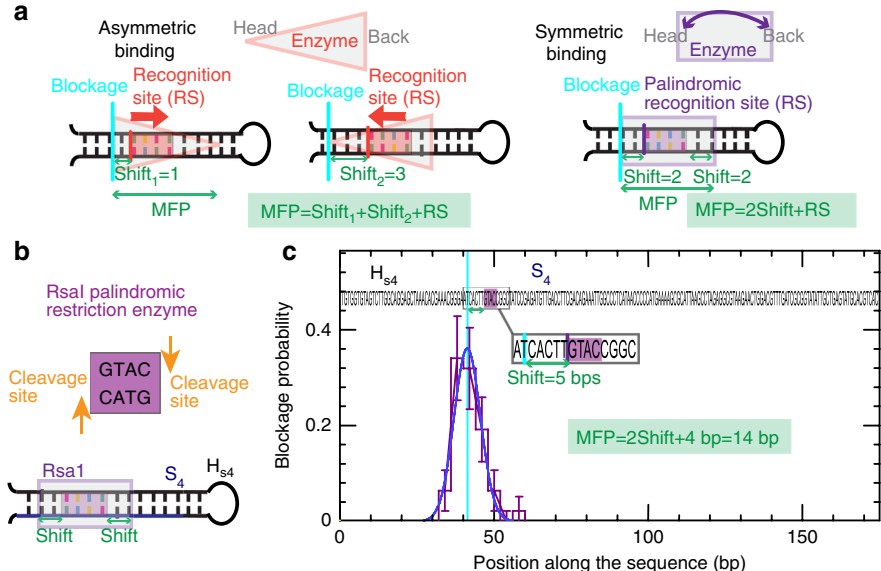

**Fig. 5** Mechanical footprint (MFP) for restriction enzymes. **a** Schematics of the binding of a large protein, such a restriction enzyme, on a DNA sequence. In the general case of asymmetric binding, depending on the enzyme orientation and on which part of the enzyme binds to the recognition sequence, the shift between the measured blockage position during hairpin unzipping (corresponding to the head or back of the enzyme) and the recognition site will be different. The MFP, corresponding to the DNA size covered by the enzyme, is computed as the sum of the shifts measured in the two enzyme orientations plus the size of the restriction site. In the particular case of symmetric binding, the central part of the enzyme binds to the recognition site, and the measured shifts from the head and back of the enzyme are expected to be the same. The MFP is then computed as twice the measured shift plus the size of the restriction site. **b** Schematics of recognition and cleavage of the palindromic Rsa1 restriction enzyme, as well as its binding to hairpins $H_{s4}$. **c** Distribution of blockage positions obtained by applying the FC protocol to the $H_{s4}$ hairpin in presence of the Rsa1 enzyme (number of beads = 72). *Error bars are inversely proportional to the square root of the number of points for each bin.* A single blockage is observed that is shifted 5 bps away from the recognition site. The MFP is estimated 14 bp, twice the shift plus the 4 bps corresponding to the restriction site

interaction sites closer to the head and back of the ligand, determining what we define as the mechanical footprint of the protein (hereafter referred to as MFP), Fig. 5a. Depending on the ligand/DNA binding map and orientation, the measured blockages will be located at different regions along the ligand, and shifted differently with respect to the recognition site. Yet, the sequence of the recognition site can be recovered with a simple correlation analysis if more than one recognition site is present along the DNA sequence (Methods section).

Here, we focus on restriction enzymes, which recognize specific DNA sequences where they bind and cleave DNA. To inhibit DNA cleavage, we perform the assays without $Mg^{2+}$, that is an essential cofactor for catalytic activity. We first tested the homodimer Rsa1, which is a type II endonuclease that recognizes and cleaves the palindromic sequences GTAC within its recognition site (Fig. 5b). The recognition sequence and cleavage sites of this enzyme have been determined from restriction endonuclease mapping using DNA fragments of known sequence and Sanger sequencing[44]. Although an extensive biophysical characterization is not available, equivalent results were found for its isoschizomer AfaI[45]. Both enzymes are known to be type IIP endonucleases, which achieve palindromic recognition by homodimer formation[46–48]. By using the FC protocol with $H_{s4}$ hairpin, which contains a single recognition site for Rsa1, we detect blockages (Supplementary Fig. 9) that are located ~5 bps ahead from this position (Fig. 5c). Due to the symmetric binding of the enzyme to the palindrome, the enzyme's head and back should be equidistant from the recognition site (Fig. 5a), leading to a MFP of 14 bps (~1.5 turns of the DNA helix), computed as twice the measured shift (5 bps) plus the recognition site (4 bps). This observation is in agreement with results obtained for EcoRI using short (34-bp) DNA hairpins, where blockage events ~6-bp ahead from the recognition sequence are observed, suggesting

that a mechanical footprint of approximately half and helical turn might be consistent between different type IIP endonucleases[23]. By using a DNA hairpin that contains two binding sites, the recognition sequence could be determined from a sequence correlation analysis (Supplementary Fig. 10).

The situation might be more complex in the general case of enzymes with non-symmetric binding, such as non-palindromic restriction enzymes, since head and back may be shifted differently with respect to the recognition site (Fig. 5a). In this situation, depending on which strand of the DNA helix the recognition site is located, the enzyme will present a different orientation and might lead to a different unzipping blockage pattern. To detect such orientation effects on ligand binding, we prepared a new hairpin, named $H_{s4s4'}$, that presents the previously designed sequence $S_4$ followed by the complementary sequence $S_4'$ (Fig. 6 and Methods) separated by a 15 nucleotide AT-rich region, that acts as spacer between the two complementary regions. In this DNA construct, for each binding site in the $S_4$ region there will be the mirror image binding site on $S_4'$, allowing to locate the head and back of the enzyme and therefore directly measure its MFP. A similar approach using independent forward and reverse hairpin constructs has been successfully used to prove asymmetric binding of protein heterodimers in DNA mismatch repair[49].

To investigate the effect of non-symmetric binding, we have tested two non-palindromic restriction enzymes: BspCNI and MnlI. These enzymes are type IIS endonucleases that have a non-palindromic recognition sequence and cleave DNA outside this recognition sequence[50, 51]. Type IIS endonucleases are typically found as monomers in solution[47, 48]. For MnlI electrophorectic mobility shift assays have determined the affinity constant of binding to DNA to be in the range 5–50 nM[52, 53], in agreement with values obtained for other type II restriction endonucleases[54].

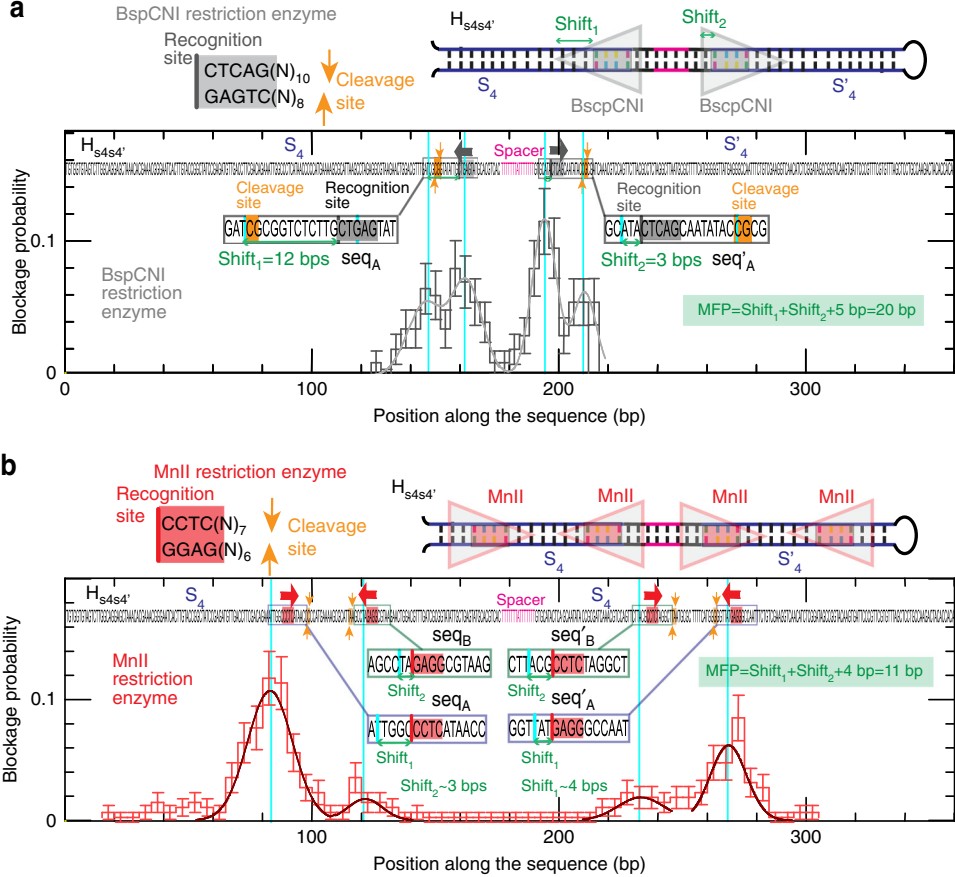

**Fig. 6** Multiple binding sites and flanking sequences in non-palindromic restriction enzyme binding. **a** Distribution of blockage positions obtained applying the FC protocol with the $H_{s4s4'}$ hairpin in presence of the BspCNI restriction enzyme (number of beads = 155). Error bars are inversely proportional to the square root of the number of points for each bin. The results show four peaks that can be related to both recognition and cleavage sites. Schematics of recognition and cleavage of the BspCNI enzyme, as well as its binding to the $H_{s4s4'}$ hairpin is shown in the upper part of the panel. The MFP is extracted from the shifts measured in the forward and reverse sequences. **b** Distribution of blockage positions obtained by applying the FC protocol to the $H_{s4s4'}$ hairpin in presence of the MnII restriction enzyme (number of beads = 139). Error bars are inversely proportional to the square root of the number of points for each bin. The results show four peaks that can be related to the four repeats of the recognition site present along the sequence. The shifts associated to the enzyme head and back, shift$_1$ and shift$_2$, can be measured in both $S_4$ and $S'_4$ regions and so the MFP. The recognition site can be also determined from the correlation analysis shown in Supplementary Fig. 10. The different weights of the fitted Gaussian functions are attributed to the different binding stabilities induced by the DNA flanking sequences. Schematics of the recognition and cleavage of the MnII enzyme as well as its binding to the $H_{s4s4'}$ hairpin is shown in the upper part of the panel

We first tested BspCNI which has the recognition site CTCAG, and cuts DNA 10 bases away from this sequence. Application of the FC protocol with $H_{s4s4'}$ and BspCNI shows blockages in two differentiated regions, one in the forward $S_4$ sequence, and the other in the reverse $S'_4$ sequence (Supplementary Fig. 11), consistent with the fact that $S_4$ contains the CTCAG motif only once. Remarkably, the distribution of blockages shows 2 peaks in each region, revealing different DNA/protein binding sites associated to a single binding event (Fig. 6a). In $S_4$, one peak is located in the recognition sequence, whereas the other one is found about 10 bases ahead, coinciding with the expected cleavage site. The reversed orientation is observed in $S'_4$ (Fig. 6a). These results show that BspCNI makes strong contacts with DNA in at least two sites: the recognition site and the cleavage site, with slightly higher affinity for the former, as deduced from the weight of the peaks. Although the recognition sequence cannot be extracted from this particular experiment (as the $S_4$ sequence contains the binding site only once), the MFP of the enzyme can be determined from the shifts measured in the forward $S_4$ and reverse $S'_4$ sequences (~20 bps). In order to investigate the effect of divalents ions on the enzyme binding, we performed experiments

in presence of $Ca^{2+}$, which in other enzymes is observed to increase affinity without inducing cleavage[55]. We find that calcium concentration increases binding affinity of the enzyme, in agreement with previous results[55], but without affecting the relative affinity of the two binding sites (Supplementary Fig. 12).

Next, we performed experiments with another non-palindromic enzyme (MnII) that has a recognition sequence (CCTC) and cuts DNA 8 bases away. As expected, blockage events are detected both in $H_{s4}$ and $H_{s4s4'}$ (Supplementary Fig. 13a, b). The distribution of blockage positions show respectively two and four peaks, one for each binding site (Supplementary Fig. 13c and Fig. 6b), and the MFP is measured to be 11 bps (~1 DNA helix turn). Note that in this case, the binding to the cleavage site does not give any clear blockage signature. This might be due to the lack of interaction between enzyme and cleavage site in the absence of $Mg^{2+}$. However note, that the peaks are very broad and they might mask some minor presence of binding on the cleavage site. The widening of the peaks is a signature observed in all restriction enzymes (Gaussian widths are from ~−4 to 9 bps, Supplementary Table 2), which might be the result of the large interaction region between

DNA and the enzymes, as compared to the small ligands presented in the first part of the work. In this case, since the recognition site appears twice in $S_4$, its sequence can be extracted from a correlation analysis (Supplementary Fig. 13d). Interestingly, we observe differences in the weight of the Gaussian functions associated to each peak (Supplementary Table 2). This asymmetry could be due to the fact that (i) we are mechanically disrupting the complex from the opposite direction, or (ii) due to the fact that different flanking sequences around the recognition site (CCTC) affect the mechanical stability of the protein-DNA complex. Measurements with hairpin $H_{s4s4'}$ allow us to discriminate between these two scenarios as this hairpin contains the forward and reverse sequence of the two binding sites embedded in different flanking context. In the first scenario (pulling orientation causes asymmetry), we would expect that the two sites pulled from the same orientation (e.g., sites 1 and 3) should have a similar height regardless of being embedded in a different sequence context. In the second scenario (flanking sequence causes asymetry), we would expect that the two sites embedded in the same flanking context (e.g., sites 1 and 4) should have a similar height, regardless of being mechanically disrupted in the opposite orientation. Experimental results show that peak weight does not relate to protein orientation but to the different flanking sequences (second hypothesis), revealing how sequence-dependent effects context can strongly affect the binding stability of the complex. Finally, we also verify the effect of $Ca^{2+}$ on the binding of MnII, finding a similar trend that for BspCNI: calcium increases binding affinity without affecting the relative affinity of the different binding sites. We also observe that calcium induces cleavage (as we quickly lose tethered beads after injection of the enzyme), in agreement with previous bulk experiments[53].

## Discussion

In this work, we present a method to determine the selectivity of small and large ligands binding to nucleic acids with near one bp resolution, based on the mechanical unzipping of DNA hairpins using a parallelized magnetic tweezers set-up. The method also allows us investigating several aspects of DNA/ligand binding kinetics and affinity. We first develop a procedure to generate DNA hairpin sequences with uniform stability and presenting all tetramer combinations of bases, that allow to obtain unbiased measurements and sequence selectivity of ligands that bind and recognize 4 DNA bps or less. These hairpins are then used to characterize the binding of very small ligands difficult to study using parallelized techniques based on fluorescent labeling or sequencing. As a proof of principle we investigate different intercalators and determine their preferred binding sites, as well as the modifications induced by flanking sequences on their selectivity. We also develop protocols to extract binding energies, the kinetic unbinding rate and the position of the transition state ($X^{\dagger}$), similarly as previously shown for DNA binding proteins using dynamic force spectroscopy[18]. We show how these measurements can provide information about the unbinding pathway as well as evidence for multiligand binding. Although the intercalators used in this work are known to uniformly coat DNA at high concentrations[21, 56, 57], binding to random non-specific sites is not observed in our conditions. Yet, we observed a widening of the binding peaks (~4-bp vs the 2-bp resolution) in particular sequence contexts, that we attribute to regions were secondary binding sites in close proximity to each other take place as well as multiligand binding. In particular we have observed this for Echinomycin and Thiocoraline in GC-rich stretches, in agreement with the known sequence-preference of these ligands[21, 38, 39]. Finally, preliminary results on ligands with

lower binding affinity, such as minor group binders, do not give clear binding signal, suggesting that the method is suited to observe molecular interactions with $K_d \leq 10\,\mu M$.

We also apply the method to study the specific binding of large proteins to DNA, which typically present different DNA binding domains (e.g., catalytic/DNA-recognition domains) that might have varying affinities. In this case, the DNA unzipping blockage analysis reveals the positions where the enzyme most strongly binds to DNA closer to its boundaries (head and back), allowing us to extract what we called the mechanical footprint (MFP) of the ligand. Although the MFP does not directly relate to the recognition site, the latter can be determined by a simple correlation analysis whenever the recognition site appears more than once along the DNA sequence. In order to measure the head and back protein anchoring sites, we have designed a hairpin that presents the DNA testing sequence followed by its reverse complement, so that enzyme binding takes place in both orientations. This also allows investigating the effects of enzyme orientation in the mechanical stability, and provides insight on site-orientation selectivity, which is important in several systems such as type III restriction enzymes, DNA mismatch repair, nucleosome dynamics or termination of DNA replication by the Tus-Ter complex[20, 37, 49, 58–60].

We show this using three different restriction enzymes, finding MFPs that range between 10 and 20 bps, in agreement with the typical size of these proteins. For two of them, we detected a single binding site per binding event, associated to the recognition site. However, for a third enzyme, multiple binding sites were detected, associated to the recognition and cleavage sites, and we determined the relative affinity between both sites. Last but not least, the effect of flanking sequences was also investigated revealing their significant contribution to the overall enzyme/DNA stability. Flanking sequences are know to play and important role on other systems, such as endonucleases[61–63] and our technique has proved to be well suited to study these effects that are difficult to measure in calorimetry studies[62].

How large are the binding motifs that can be identified with this method? As a proof of principle, we show that a sequence with a flat free energy landscape and containing all possible tetramers ($K = 4$, ~256 combinations) can be embedded in a 170 bp hairpin for single-molecule footprinting. We chose 4-mers as it is an appropriate size to study small ligands within its flanking context and sufficiently representative of most DNA endonucleases, although the hairpin synthesis could be easily extended to longer motifs. There are two main aspects to take into account for that: (1) The length of the hairpin must be increased to accommodate all possible k-mer sequences (e.g., $K = 7$ requires 16,384 combinations, $K = 8$ requires 65,536 combinations). Typical DNA unzipping experiments can accommodate hairpins up to 10 kb, although experiments with hairpins up to 50 kb have also been performed[29, 30]. Taking into account that hairpins shorter than the total combinatorial length can be achieved due to the complementary Watson-Crick symmetry between the forward and reverse strand of the hairpin (e.g. AAAA is equal to TTTT), experiments with hairpins containing all possible 7-mers are within experimental reach. Longer k-mer combinations can be achieved by partitioning the sequences between a few DNA hairpins. (2) The roughness of the free energy landscape must remain within the order of thermal fluctuations ($1–4\,k_BT$) to obtain unbiased results and be able to extract kinetic and thermodynamic parameters as shown for echinomycin. This becomes challenging for longer k-mers, as hairpins have to accommodate longer AT-rich and GC-rich motifs. The combination of very stable motifs (GCGCGCGC) and very weak motifs (e.g., TTTTTTT) should increase the roughness of the free energy landscape. A way to obtain unbiased results for

longer k-mer motifs is to partition the sequences between two or several hairpins with different levels of GC-content. Otherwise, experiments could also be performed in a single hairpin that contains regions of different thermodynamic stability and using the blocking oligo approach to study these different regions sequentially in a single experiment (by applying an equivalent $\Delta F$ test force in each region). Finally for very long DNA binding motifs, such as those shown for some transcription factors (up to 15 bp), it might be more informative to perform a preselection of target binding sites to reduce the size of the combinatorial space being able to study this subset of target binding sequences within different flanking contexts.

Force-spectroscopy techniques have been used to study the non-specific binding of small ligands to DNA by following changes in the elastic properties of long DNA molecules in stretching experiments[21, 22, 56, 57, 64–66]. On the other hand, DNA unzipping assays have been used to observe the binding position of large proteins in particular biological contexts[17, 20, 37]. However, to the best of our knowledge, this is the first work that proposes a general unbiased method to study the binding selectivity of small ligands and large proteins against a large number of sequences using the mechanical unzipping of DNA hairpins. The method developed here, that applies for magnetic and optical tweezers, could be used in other force-spectroscopy parallelized set-ups, such as acoustic trapping and nanophotonic optical traps[67, 68], for high-throughput screening of molecular interactions. Yet, an unexplored advantage of the proposed single-molecule footprinting technique is the possibility to study interactions involving more complex topological configurations, such as DNA-protein loop formation (e.g., transcription factors, chromatin insulator proteins) or ternary complexes between small ligands and enzymes binding DNA.

# Methods

**Experimental set-up**. Magnetic tweezers experiments are performed with a PicoTwist magnetic tweezers instrument (www.picotwist.com). DNA molecules are tethered between a glass surface treated with anti-digoxigenin antibody (Roche) and a 1-µm streptavidin-coated Dynal magnetic bead (Invitrogen). DNA molecules are manipulated and stretched by capturing the bead in a magnetic trap generated by a pair of permanent magnets. A 100× 1.25 N.A. microscope objective (Olympus) images the bead onto a CCD camera for real-time position 3D tracking at 30 Hz. The image of the bead displays diffraction rings that are used to estimate its 3D position and the applied force by using a calibration curve[28].

**Differential measurement of the molecular extension**. In order to measure the molecular extension of a given hairpin, we measure the extension of the bead where the hairpin is tethered with respect to a fixed bead (attached non-specifically to the glass surface). This differential measurement allows to effectively reduce drift effects[28]. Moreover, in order to achieve a better alignment between different cycles and extract the plateaus generated by the different blockages, for each bead, we subtract to the molecular extension during phase 2 ($F_{test}$ phase) the average value of the molecular extension in the phase 3 ($F_{high}$ phase). This operation leads to a good collapse of the data from different cycles (Supplementary Fig. 3). From this data, we compute the histograms of molecular extension and extract the position of the blockages (Fig. 2b).

**Free energy landscape**. The stability of a given DNA sequence is roughly determined by the content of GC vs AT bps, GC bps being about twice more stable than AT bps in average (the actual bp energy also depends on the stacking energy with the neighboring bases). When a high enough force is applied at the extremities of a DNA hairpin, the DNA unzips sequentially. For an $N$ bps hairpin the free energy of the different partially unzipped DNA conformations, with the first $n$ sequential bps unwound and the next $N − n$ bps formed, at the force $F$, can be written as: $G_F(n) = \Delta G^0(n) − n \cdot G^{ssDNA}(F) + G_{loop}$, where $\Delta G^0(n)$ is the free energy associated to the $(N − n)$ final bps of the DNA hairpin at zero force, $G^{ssDNA}(F)$ is the free energy associated to the stretching of the two nucleotides of ssDNA released in unzipping a single bp at force $F$, and $G_{loop}$ corresponds to the free energy associated to the loop formation. The free energy landscape $G_F(n)$ at a given force $F$ exhibits, in general, a rough free energy landscape, presenting many barriers associated to the specific sequence. For an $N$ bps DNA random sequence the height of the largest dominant barrier increases with the square root of $N$, reaching 10 $k_B T$

for $N = 100$ bp[69]. In particular, a barrier (and the associated transition state) is generated when AT-rich regions are alternated with GC-rich regions.

Cooperative two-state force unfolding and refolding is observed whenever barriers, which could generate intermediates in the unzipping pathway, are not too large. Therefore, by switching the force between a low value ($F_{low}$, hairpin formed) and a higher value ($F_{test} > F_c$, hairpin unfolded), we can induce the cooperative unzipping and rezipping of the hairpin (Fig. 1d). Working with two-state hairpins with a flat free energy landscape ensures that, in presence of DNA ligands, the observed blocking events during DNA unzipping are exclusively due to the extra stabilization energy brought by the bound ligands, and not by the DNA sequence itself (e.g., kinetic barriers due to CG-rich regions).

**DNA hairpin synthesis**. $H_0$, $H_{s4}$, and $H_{s4s4'}$ hairpins are synthesized with the protocol described in ref. [21]. A plasmid that contains the sequence of interest is embedded between the restriction sites of Tsp45I and TspRI. This region of the plasmid is polymerase chain reaction amplified and the product is digested with the two enzymes. A set of oligonucleotides is designed to create the handles structure on the TspRI-digested end. Similarly, an oligonucleotide complementary to the Tsp45I end that folds into the end-loop structure is annealed and ligated to create the final hairpin structure. A final gel purification step is added to remove competing structures. Supplementary Fig. 2 shows the schematics of the 3 constructs, and the set of oligonucleotides used is specified in Supplementary Table 1.

**Ligands and buffer conditions**. The experiments were conducted at a temperature of 25 °C. All experiments with intercalators and minor group binders were performed in TE buffer (Tris 10 mM, ethylenediaminetetraacetic acid 1 mM) pH 7.5, 100 mM NaCl, 0.01% NaN$_3$ and DMSO. Echinomycin was purchased from Merck and Actinomycin, DAPI and netropsin from Sigma-Aldrich. Thiocoraline was provided by Pharmamar. Concentrations used ranged from 100 nM to 10 µM. The different restriction enzymes were purchased from New England biolabs and used as 1:100 dilutions of the original stock: RsaI (10,000 U/ml, 50 nM dimer [1.7 µg/ml], MW: 19 kDa monomer), BspCNI (2000 U/ml, 2 µM [270 µg/ml], MW: 105 kDa), MnlI (5000 U/ml, 1.2 µM dimer [90 µg/ml], MW: 38 kDa monomer).

**Kramer's Bell-Evans theory for force-dependent unbinding**. In the 1D Kramer's description, the ligand unbinding reaction can be seen as the one dimensional diffusion of particle over a barrier B (Fig. 2d). The barrier B, corresponding to the maximum in the free energy landscape, is located at a distance $X^+$ from the ligand bound state, along the reaction coordinate axis (the end to end distance of the hairpin DNA molecule along the z-force axis, $Z_e$). In this two-state description, the height of the barrier B decreases linearly with the applied force, $F$, and the distance of the bound state to the transition state, $X^+$; and the rate of escape over the barrier, $k$, shows an exponential dependence on the applied force $F$: $k = k_0 \exp(FX^+/k_BT)$. By measuring the dependence of the average block lifetime, $<\tau> = 1/k$, on the applied force, and representing $\ln[k](F)$ we can extract $k_0$ and $X^†$ from the lineal fits: $\ln[k] = \ln[k_0] + FX^†/k_BT$ (Fig. 2e).

**Measuring binding energy from equilibrium hopping traces**. From the equilibrium hopping traces obtained with $H_0$ at fixed force $F_{hop}$ (Fig. 3a), we can compute the probability of populating each partially unzipped configuration, $p_i$ ($i = 1,2,3$), as follows. We compute the distribution of molecular extension at $F_{hop}$ and fit three Gaussian functions, corresponding to the three partially unzipped hairpin configurations blocked at one of the binding sites (Fig. 2f, g and Supplementary Table 2). The probability $p_i$ is then computed as the ratio between the weights of the Gaussian fits $A_i$, as $p_i = A_i/(A_1 + A_2 + A_3)$. If we assume that each block is generated by the presence of a single Echinomycin blocking the full unzipping, and that Echinomycin is bound to the preferred GpC sites but not to the other sites, we can then write the difference in free energy between the partially unzipped states, $j$ and $i$, at force $F_{hop} = F$ as: $\Delta G_F(i, j) = G_F(i) − G_F(j) = k_BT \ln[p_j/p_i] = \Delta\Delta G^0_{i,j} − \Delta G^{ssDNA}_{i,j}(F) + \Delta G_{int}$, where $\Delta\Delta G^0_{i,j}$ is the free energy at zero force associated to the stretch of $n_i − n_j$ bps (with $n_i$ and $n_j$ being the number of bps formed in states $i$ and $j$); $\Delta G^{ssDNA}_{i,j}(F)$ is the stretching free energy associated to the $n_i − n_j$ unzipped bps and can be written as $\Delta G^{ssDNA}_{i,}(F) = (n_i − n_j) \cdot G^{ssDNA}(F)$, where $G^{ssDNA}(F)$ is the free energy associated to the stretching of the two nucleotides of ssDNA released in unzipping a single bp at force $F$; and $\Delta G_{int}$ is the binding energy of Echinomycin to a GpC site. The value of $\Delta\Delta G^0_{i,j}$ is estimated using the nearest neighbor free energies from Mfold[70] and the stretching free energy $G^{ssDNA}(F)$ using the freely joined chain model. This analysis gives an estimation of the specific binding energy of Echinomycin of ~7 $k_B T$ at 300 nM Echinomycin concentration (Supplementary Table 3).

**Errors and bias in the determination of binding sites**. Large differences in the number of times the different sites appear in the DNA sequence tested induce bias in the determination of preferred ligand binding sites. For instance, hairpin $H_0$ presents a single CpC step very close to one of the CpG steps, whereas the other dinucleotides appear several times distributed along the sequence (e.g., 20

repetition of ApT steps). This leads to errors and bias in the probability of observing blockages at the CpC step (due to the presence of neighboring sites) and can lead to misinterpretation. The three studied intercalators with $H_0$ present the CpC as a second preferred binding site (Fig. 4b). However, the comparison with results obtained with hairpin $H_{s4}$ (Fig. 5d) reveals that this is an artifact caused by the under-representation of this dinucleotide in hairpin $H_0$, a fact that is also reflected in the large error associated to this dinucleotide measurement (Fig. 4b).

**Simulations**. Monte Carlo simulations to generate sequences that have all combination of $K$ bases at least once and not more than twice were performed using simulated annealing as follows. We initially generate random sequences of $N$ bases and select those that include all combinations of $K$ bases at least once and not more than twice. We next propose one-base changes, which conserve all combinations of $K$ bases and accept them if they decrease the roughness of the free energy landscape, defined as $R = \sum_{j=1}^{N} \left( \sum_{i=1}^{i<j} \alpha_i - j\alpha_{mean} \right)^2$, where $\alpha_i$ is the nearest neighbor free energy associated to the bp $i$ and $\alpha_{mean} = \sum_{i=1}^{i=N} \alpha_i / N$. Minimizing the roughness $R$ is a way of minimizing the height of the barriers in the unzipping free energy profile of the DNA sequence and consequently flattening the energy profile $G_F(n)$. We focus on the study of sequences that includes all tetra-nucleotide combinations ($K = 4$), with a fixed length of 170 bps. The roughness minimization leads to sequences presenting flat profiles with small barriers whose height are on the order of thermal forces ($1$–$3\ k_BT$), Fig. 5b. Such small barriers can be easily overcome by Brownian fluctuations and are not expected to generate any detectable block to the unzipping process. In Supplementary Fig. 7 are shown the results for the roughness of the free energy landscape $R$ as a function of the Monte Carlo steps for a sequence of $N = 170$ bps that includes all tetranucleotide combinations ($K = 4$). As expected the roughness $R$ decreases as the number of Monte Carlo steps increases.

**Correlation analysis to determine the recognition sequence**. For any ligand at least two repetitions of the recognition sequence along the DNA tested are required to determine the ligand recognition sequence. In order to determine the recognition sequence, we perform a simple correlation analysis. We compare the DNA regions for stretches of $K = 4$ bps along windows of a length of 10 bps at right and left of the blockage positions and find the sequence with maximum similarity. For this, we compute the correlation functions $C_{i,j}$ and $C'_{i,j}$ between points along the DNA sequence $i$ and $j$ as: $C_{i,j} = \sum_{k=0,3} c_{i+k,j+k}$ and $C'_{i,j} = \sum_{k=0,3} c'_{i+k,j+3-k}$, where $c_{i+k,j+k} = 1$ if the base at position $i + k$ and $j + k$ are the same and zero otherwise, and $c'_{i+k,j+3-k} = 1$ if the base at position $i + k$ and $j + 3 - k$ are the complementary (A vs T and C vs G) and zero otherwise. We next define the similarity function $S_{i,j}$ for each couple $(i, j)$ as the maximum between $C_{i,j}$ and $C'_{i,j}$, $S_{i,j} = \text{Max}\{C_{i,j}, C'_{i,j}\}$. $S_{i,j}$ can take values from 0 to 4 depending on how close are the sequences $(i, i + 1, i + 2, i + 3)$ and $(j, j + 1, j + 2, j + 3)$, the maximum value of 4 corresponding to the situation where the 4 bases coincide or are complementary. The recognition sequence can be identified as the sequence $(i, i + 1, i + 2, +3)$ verifying $S_{i,j} = 4$. The results for the Rsa1 and MnII enzymes are depicted in Supplementary Figs. 10d, e and 13d.

**Data availability**. All relevant data are available from the authors.

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

### Acknowledgements

All authors acknowledge funding from ERC grant MagReps 267 862, FP7 grant Infernos 308850, and Icrea Academia 2013.

### Author contributions

M.M. conducted Magnetic tweezers assays and performed the data analysis. J.C.-S. carried out Optical tweezers assays and prepared the DNA substrates. M.M., J.C.-S., V.C. and F.R. designed experiments and wrote the paper.

### Additional information

**Competing interests:** The authors declare no competing financial interests.

