## [Peer Review File · Nature Communications]

Reviewers' comments:

Reviewer #1, an expert in magnetic tweezers (Remarks to the Author):

Referee Report for "Single molecule high-throughput footprinting of small and large DNA ligands" by Manosas et al.

In the manuscript Manosas et al. use DNA hairpins in magnetic tweezers to detect and quantify the binding of DNA binding small-molecules and proteins. Their elegant assay enables determination of the binding sequences and footprints as well as binding kinetics. Overall, I find the work well executed and of significant interest. This assay has the potential to expand the repertoire of DNA binding assays by a sensitive and quantitative new tool.

However, I do think that the work could be further strengthened and clarified by addressing the points listed below.

--- Footprinting data can, at least in principle, have single base resolution (and not just "of a few base pairs", page 2). See e.g. classic work by

Hydroxyl radical "footprinting": high-resolution information about DNA-protein contacts and application to lambda repressor and Cro protein.

Tullius TD, Dombroski BA.

Proc Natl Acad Sci U S A. 1986 Aug;83(15):5469-73.

In addition: Are there footprinting data available for any of the proteins investigated in this present manuscript? It might be of interest to quantitatively compare the different methods?

--- The concept of using DNA hairpins to sense binding is not completely novel. It would be good to at least briefly mention and cite several earlier studies in the same direction:

Single-molecule methods for ligand counting: linking ion uptake to DNA hairpin folding.

Dittmore A, Landy J, Molzon AA, Saleh OA.

J Am Chem Soc. 2014 Apr 23;136(16):5974-80. doi: 10.1021/ja500094z.

used a hairpin in magnetic tweezers to detect ion binding to DNA.

A molecular tuning fork in single-molecule mechanochemical sensing.

Mandal S, Koirala D, Selvam S, Ghimire C, Mao H.

Angew Chem Int Ed Engl. 2015 Jun 22;54(26):7607-11. doi: 10.1002/anie.201502580.

used a functionalized hairpin for antibody detection.

High Spatiotemporal-Resolution Magnetic Tweezers: Calibration and Applications for DNA Dynamics.

Dulin D, Cui TJ, Cnossen J, Docter MW, Lipfert J, Dekker NH.

Biophys J. 2015 Nov 17;109(10):2113-25. doi: 10.1016/j.bpj.2015.10.018.

used a hairpin to detect binding to ssDNA, in essence the reverse of the strategy used in the present manuscript.

--- During the "ligand probing cycle" a "high" force of 25 pN is applied and it is assumed that this removes the bound ligand. However, is 25 pN really enough to completely remove associated the ligands?

--- The authors discuss at some length their procedure to design a hairpin with a flat energy landscape and an equal distribution of binding sites, which is a very useful tool to optimize their assay. However, it appears to me that such a design will quickly become unfeasible for larger binding sites? It might be good to discuss these limitations in some more detail.

Minor points:

- Page 3: Insert space between number and unit at "1nm".
- Page 6, top: "we found out a ..." better: "we found a..."
- Page 13: "25 C" better: "25°C"

Reviewer #2, an expert in DNA binding proteins (Remarks to the Author):

In this report, Manosas et al. provide a single-molecule approach that uses magnetic tweezers and DNA hairpin unfolding dynamics to characterize the binding sequence, affinity and kinetics of small molecule binding such as DNA intercalators. They apply the same method to describe the DNA binding sequence of large DNA binding proteins such as restriction enzymes. The assay used here is well described and provides high quality data, as the authors have already demonstrated in previous studies. The authors also performed a sound analysis to characterize the DNA binding sequence of the different molecules studied here. Overall, this is a nice biophysical work.

However, hairpin opening for molecule binding detection has been used in many previous articles, with either high throughput magnetic tweezers or optical tweezers. In addition to the cited literature, there is also: Shundrovsky et al. *Nat. Struct. Mol. Bio.* 2006, Hall et al. *Nat. Struct. Mol. Bio.* 2009, Jin et al. *Nat. Struct. Mol. Bio.* 2010, Dame et al. *ChemBioChem* 2013, Dittmore et al., *JACS* 2014, Mandal et al. *Angewandte* 2015, Dulin et al. *Biophys. J* 2015. This method is clearly not new in that respect. High-throughput magnetic tweezers (see my comment below) have also been recently used with DNA hairpins to characterize the mechanism of the Tus-Ter interaction (Berghuis et al. *Nat. Chem. Bio.* 2015). Therefore, the methodology used here, though well performed, is not new.

Furthermore, this method has not been applied here to answer a biological question of great importance.

In conclusion, I think that this work is not appropriate for publication in *Nature Communications*, according to its broad scientific audience. I believe this work is better suited for a more specialized journal, e.g. *Biophysical Journal*.

If the editorial board decides to proceed with the paper, the following points need to be addressed.

1. I understand that adding "high-throughput" in the title sounds "hot", but the authors do not show any evidence for a high-throughput assay: I could not find a graph combining all the acquired traces from a single experiment, or an image of the field of view during a typical experiments, that clearly shows hundreds of tethered beads and few reference beads. Furthermore, the statistical error represented by the error bars in the histograms and in the bar plots seem relatively large (Fig. 3b, Fig. 5c, Fig. 6a, Fig. 6b), underlying not so large data sets. Therefore, if the authors do not provide more evidence, the mention "high-throughput" must be removed from the title. The authors should also have cited publications in the field that have already demonstrated high-throughput force spectroscopy measurements, e.g. de Vlaminck et al. *NanoLetter* 2011 and *PlosOne* 2012, Dulin et al. *NAR* and *Cell Rep* 2015, Duderstadt et al. *Mol Cell* 2016... The most recent studies show ~100s to close to 1000 tethered followed simultaneously with a similar spatiotemporal resolution. I believe that the authors should demonstrate such capabilities to maintain the "high-throughput" qualification in their title.
2. The study of Echinomycin is well done and well described, with a detailed and clear analysis. Given that the authors are using a hairpin to detect molecule binding, they should cite the following work: Dittmore et al., *JACS* 2014, Mandal et al. *Angewandte* 2015, Dulin et al. *Biophys. J* 2015.
3. I could not find any mention of the number of data points per histogram or bar plot represented in this study. The author must provide the statistics for each experiment performed in the study. I

recommend them to add a table in the Supplementary Information resuming all the conditions and their respective statistics. A clear description of the methodology for the extraction of the plateau is lacking, as well as how the error bars in the bar plots and the histograms are derived.

4. Little is mentioned about the restriction enzymes binding conditions and affinity. Given that half of the presented work reports on the detection of bound restriction enzymes, it is necessary to say something about their biochemistry, e.g. to which type belongs the studied restriction enzymes, their respective K_d in the buffer used here, their mechanism of binding and cleaving... I have not been able to find any citation of previous biochemical work on Rsa1 for example. The authors mentioned that Rsa1 is a homodimer that binds symmetrically, head and back being equidistant from the recognition site, and have measured a footprint of ~ 14 bps (l. 200, p.8), but I have found neither citation nor complementary biochemical studies supporting their results or assertion. The same works for the other enzyme. I recommend that the authors include reference(s) that support(s) better their work. If the authors cannot provide supporting literature, they should provide bulk biochemical assays to support their claims. I strongly recommend the authors to add in their study a characterization of a well-described restriction enzyme, e.g. EcoRV (also provided by NEB), which will provide them with plenty of literature to compare their results with. This would allow them to validate their mechanical footprinting method.

5. The authors look at the binding affinity of enzymes, but they did not provide any meaningful concentration (only Units/ml). I recommend them to contact NEB, as they purchased all their enzymes from that company, to obtain the concentration of the stocks. This would be meaningful to compare with the K_d on random and specific DNA binding sequence.

6. All the enzymatic study performed here use a buffer containing EDTA (TE buffer, see Methods), which traps any residual traces of divalent ions in the solution. I understand that the authors do not want any DNA cleavage during their experiments. However, the binding affinity and specificity of restriction enzymes is strongly dependent on the presence of divalent ions, e.g. EcoRV. Though the presence of Mg will generate DNA cleavage, and is therefore incompatible with the study presented here, the authors can use other divalent ions to conserve the binding affinity observed in the presence of Mg while impairing DNA cleavage. They could look for example at the work published by Bowen and Dupureur, *Biochemistry* 2003 to find such a divalent ion.

7. On BspCNI, the authors observe two binding sites per binding region (Fig. 6b), and propose that one comes from the recognition site, the other one from the cleavage site. Do they have any biochemical proof to bring such conclusion? Is there anything in the literature about it? Did they probe whether this could originate from non-specific DNA binding of the enzyme? I believe that the authors should provide experiments with at least another concentration of BspCNI to demonstrate that it is not non-specific binding. Adding a divalent ion that promotes binding but impair cleavage would help to show that it is the cleavage site that binds.

8. To definitely demonstrate the mechanical footprint of BspCNI, I recommend the authors to perform an experiment similar to the one described in Fig. 6a, but with the second restriction site inverted, to determine whether the distance between the two Gaussian peaks in Fig. S8c varies. The first two peaks (Fig. 6a, ~ 150 and 160 bp) representing the binding on the first site should remain unchanged, providing an absolute calibration in position, when the second should demonstrate the asymmetry of the binding of BspCNI. The same works for MnII. This will further demonstrate the quality of their calibration.

9. In the last paragraph of the Results section (l.239-246), the authors rightly highlight a difference in the weight of each Gaussian from Fig. 6b and Fig. S10c using MnII. They explain that they can discriminate the origin of this difference from the similar binding region and flanking sequences that are inverted in the hairpin S4. As they use the same sequence, I would conclude that the origin of the difference in weight comes from the protein orientation. However, they conclude that the difference comes from the flanking sequence. This needs to be rewritten because it is unclear what the authors conclude here.

10. The authors should provide a statistical analysis of the restriction enzyme blocking time, as they have performed for the Echinomycin. This will inform the reader about the mechanism of release of the enzyme on the DNA sequence upon DNA unzipping.

11. The description of the correlation analysis to determine the recognition sequence in the Methods section is concisely and well explained but hard to understand without drawing it. I think

that it is necessary for the authors to include a schematic describing this analysis in the Supplementary Information.

12. The results show here the binding DNA sequence for well-known restriction enzymes (they are referenced on Wikipedia). How does this method perform with enzymes that have an unknown binding sequence? It is necessary to add such a study to fully demonstrate the generality of this method.

Reviewer #3, an expert in single molecule biophysics (Remarks to the Author):

In this manuscript, Manosas et al. describe a method for mapping bound ligands by unzipping DNA using magnetic tweezers that can simultaneously monitor multiple tethered beads. Although DNA unzipping for mapping methods has been previously demonstrated, this manuscript distinguishes itself by systematically designing DNA sequences for constant force experiments and providing detailed studies of several classes of ligands and binding proteins. I would like to recommend publication in Nature Communications if the authors are able to fully address the following comments.

1. It seems that the authors might not be aware of a relevant publication for DNA foot-printing using unzipping DNA (Jiang et al., Detection of high-affinity and sliding clamp modes for MSH2-MSH6 by single-molecule unzipping force analysis. *Mol Cell*, 2005, 20:771-81). Jiang et al. mapped the footprint of the DNA repair enzyme by unzipping a single DNA molecule from both the forward and reverse directions, similar to the current manuscript (figures 5 and 6). Please include a discussion of comparison with this work.

2. There is another relevant publication that also seems to have been overlooked (Koch et al., Dynamic force spectroscopy of protein-DNA interactions by unzipping DNA double helix. *Phys. Rev. Lett.* 91:028103, 2003). Koch et al. determined the equilibrium binding constant of a bound protein K_d , the rate of dissociation k_{off} , and the distance of the bound state to the activation barrier by using DNA unzipping dynamic force spectroscopy. Again, please a discussion of comparison with this work.

3. The authors stated that the magnetic tweezers instrument provides a resolution of ~ 2 bp. However, data in figure 2a suggest significant drift over time, and figure 2b also shows peak widths that are much broader than 2 bp. Please show evidence (which can be included in the SI) for how this 2 bp resolution is determined. Related to this, the DNA attachment point on a magnetic bead varies from bead to bead. How is the absolute extension determined? Please include a discussion under Methods.

4. The theory of dynamic force spectroscopy was used to find k_0 , which was obtained as the rate at zero force of unzipping. However, the unzipping force itself is ~ 15 pN, below which force no longer has any impact on the lifetime of a bound ligand. Therefore, it would seem that k_0 should not be obtained at force = 0, and instead at force = 15 pN. What is the reason to extrapolate the force to zero?

5. The authors wrote in lines 133-134: "Interestingly, the unbinding rate associated to the last binding site corresponds to the product of unbinding rates for single ligands, suggesting that the binding of two contiguous Echinomycin bis-intercalators is cooperative." This does not seem to make sense based on unit analysis. The unbinding rate has units of $1/s$, and thus the product of two rates would have units of $1/s^2$. These two quantities cannot be the same. Please clarify.

6. For Figure 4c, the detected peaks are not located at the same locations relative to the XCGY sites. What is the cause for this misalignment?

REVIEWER'S RESPONSE

Following, there are the detailed answers to the 3 reviewers. Their comments and questions are highlighted in bold. Amendments to the manuscript that comply with reviewers' demands have been introduced in the main text and the Supplementary Information and are listed in the accompanying documents. All calls to figures, tables, equations, references of the main text and supplementary material refer to the new revised version if not explicitly said otherwise. Page and line numbering correspond to the version with changes tracked in red.

Peer Reviewer #1:

In the manuscript Manosas et al. use DNA hairpins in magnetic tweezers to detect and quantify the binding of DNA binding small-molecules and proteins. Their elegant assay enables determination of the binding sequences and footprints as well as binding kinetics. Overall, I find the work well executed and of significant interest. This assay has the potential to expand the repertoire of DNA binding assays by a sensitive and quantitative new tool.

However, I do think that the work could be further strengthened and clarified by addressing the points listed below.

1) Footprinting data can, at least in principle, have single base resolution (and not just "of a few base pairs", page 2). See e.g. classic work by Hydroxyl radical "footprinting": high-resolution information about DNA-protein contacts and application to lambda repressor and Cro protein. Tullius TD, Dombroski BA. Proc Natl Acad Sci U S A. 1986 Aug;83(15):5469-73.

We thank the reviewer for providing this reference. In the revised version of the manuscript we have included the reference of Tullius et al. and corrected the text that now reads:

"In DNA footprinting the binding sequence and coverage size of a ligand are determined from the restriction pattern of a radiolabelled DNA molecule that has been incubated with the ligand, achieving up to one base-pair (bp) resolution."

We have also included two additional references on high-resolution hydroxyl footprinting [1,2].

[1] Cons, Benjamin MG, and Keith R. Fox. "High resolution hydroxy radical footprinting of the binding of mithramycin and related antibiotics to DNA." *Nucleic acids research* 17.14 (1989): 5447-5460.

[2] Jain SS, Tullius TD. Footprinting protein-DNA complexes using the hydroxyl radical
Nature Protocols 3, 1092-1100 (2008)

2) In addition: Are there footprinting data available for any of the proteins investigated in this present manuscript? It might be of interest to quantitatively compare the different methods?

We could not find references describing DNA footprinting experiments with the restriction enzymes used in this work (RsaI, BspCNI or MnlI) or its isoschizomers. The sequence recognition pattern for these enzymes has been determined from restriction endonuclease mapping using DNA fragments of known sequence (lambda-phage, ϕ X174, SV40, M13mp18 and plasmids pBR322 and pUC18) [1,2,3]. The cleavage site was determined from Sanger sequencing experiments of digested versus undigested fragments [1,2,3]. We have added the references on the biochemical characterization of these enzymes in the discussion of the results. This information has been added to the discussion of the results in lines 237-274 of page 9 and 10.

[1] Lynn, S. P., et al. "RsaI: a new sequence-specific endonuclease activity from *Rhodospseudomonas sphaeroides*." *Journal of bacteriology* 142.2 (1980): 380-383.

[2] Jurenaite-Urbanaviciene, S., et al. "Characterization of BseMII, a new type IV restriction-modification system, which recognizes the pentanucleotide sequence 5'-CTCAG (N) 10/8." *Nucleic acids research* 29.4 (2001): 895-903.

[3] Kamp, D., et al. "Mapping of restriction sites in the attachment site region of bacteriophage lambda." *Molecular and General Genetics MGG* 154.3 (1977): 231-248.

3) The concept of using DNA hairpins to sense binding is not completely novel. It would be good to at least briefly mention and cite several earlier studies in the same direction:

Single-molecule methods for ligand counting: linking ion uptake to DNA hairpin folding. Dittmore A, Landy J, Molzon AA, Saleh OA. J Am Chem Soc. 2014 Apr 23;136(16):5974-80. doi: 10.1021/ja500094z. Used a hairpin in magnetic tweezers to detect ion binding to DNA.

A molecular tuning fork in single-molecule mechanochemical sensing. Mandal S, Koirala D, Selvam S, Ghimire C, Mao H. Angew Chem Int Ed Engl. 2015 Jun 22;54(26):7607-11. doi: 10.1002/anie.201502580. Used a functionalized hairpin for antibody detection.

High Spatiotemporal-Resolution Magnetic Tweezers: Calibration and Applications for DNA Dynamics. Dulin D, Cui TJ, Cnossen J, Docter MW,

Lipfert J, Dekker NH. *Biophys J*. 2015 Nov 17;109(10):2113-25. doi: 10.1016/j.bpj.2015.10.018. Used a hairpin to detect binding to ssDNA, in essence the reverse of the strategy used in the present manuscript.

We thank the reviewer for providing these key references that have helped us correct the previous underrepresentation of relevant force-spectroscopy work using DNA hairpins to detect binding. We have included these references in the introduction and discussion of the results.

4) During the “ligand probing cycle” a “high” force of 25 pN is applied and it is assumed that this removes the bound ligand. However, is 25 pN really enough to completely remove associated the ligands?

In our setup 25 pN is the largest force we can reach with the 1 micron beads we are using. We use this force ($F_{\text{high}}=25$ pN) to remove the ligand if it is still bound after the test phase (phase 3 in the force cycle). The removal of the ligand is experimentally observed as a sudden increase in the molecular extension, recovering the maximal extension (Z_{open}) corresponding to the fully opened hairpin at 25 pN (e.g Fig. 1e). For all ligands studied in this paper, this force is enough to remove the ligand from the DNA in most cycles. If the ligand remain for some cycles, these events can then be detected and treated accordingly. However, it might be possible that, for particular buffer conditions, some ligands have larger affinities and require larger forces to be released from DNA. In such a case, one can first try to increase the duration of the high force phase and, if this is not sufficient to unbind the ligand, one could still use the proposed approach but using larger DNA beads (e.g. 2.8 μM beads), which allow reaching larger forces. (Force are typically 8 to 10 times bigger with 2.8 microns beads). Finally let us add that optical tweezers studies (capable of reaching forces higher than 25 pN) of the kinetics of some of these ligands binding to short DNA hairpins show that forces of 25 pN are large enough to remove the ligand from the DNA (e.g. see Figures 1B and 3A in J. Camunas-Soler, A. Alemany and F. Ritort, *Experimental measurement of binding energy, selectivity, and allostery using fluctuation theorems*, *Science*, **355**, Issue 6323, 412-415 (2017))

We have added the following discussion in the results section (page 5 line 111):

“For all ligands studied in this paper, the force $F_{\text{high}} \sim 25$ pN is enough to force-unbind the ligand from the hairpin in most cycles. This fact is supported by optical tweezers studies of the kinetics of some of these ligands binding to short DNA hairpins [1]. For ligands showing stronger affinities to the hairpin, the protocol can be modified by applying either (i) a longer F_{high} step, or (ii) a higher force (up to 100 pN using 2.8 μm beads). A similar approach can be used to increase binding lifetime statistics in the F_{test} step.”

[1] J. Camunas-Soler, A. Alemany and F. Ritort, *Experimental measurement of binding energy, selectivity, and allostery using fluctuation theorems*, *Science*, **355**, Issue 6323, 412-415 (2017)

5) The authors discuss at some length their procedure to design a hairpin with a flat energy landscape and an equal distribution of binding sites, which is a very useful tool to optimize their assay. However, it appears to

me that such a design will quickly become unfeasible for larger binding sites? It might be good to discuss these limitations in some more detail.

As the referee highlights the design of DNA hairpins with a flat free energy landscape and equal distribution of binding sites is a key aspect of the force-spectroscopy footprinting technique. In this work we provide DNA hairpin sequences to study all possible tetramer combinations (K=4, 256 combinations) in a single hairpin 170-bp long. We chose 4-mers as this is an appropriate size to study small ligands within its flanking context (as they typically have recognition sequences <4 bp), and sufficiently representative of most DNA endonucleases.

There are two main aspects to take into account in the design of hairpins containing longer K-mer combinations for force-spectroscopy footprinting experiments (e.g. to study transcription factor binding):

1) The length of the DNA hairpin must be increased to accommodate all possible k-mer sequences (e.g. K=7 requires 16,384 combinations, K=8 requires 65,536 combinations and K=9 requires 262,144). Typical DNA unzipping experiments can accommodate hairpins up to 10 kb, although experiments with hairpins up to 50 kb have also been performed [1,2]. From this perspective, and taking into account that hairpins shorter than the total combinatorial length can be achieved due to the complementary Watson-Crick symmetry between the forward and reverse strands of the hairpin (e.g. AAAA is equal to TTTT), we think that experiments with hairpins containing all possible 7-mers are within experimental reach. Longer k-mer combinations could be achieved by partitioning the sequences between a few DNA hairpins.

2) The roughness of the free-energy landscape should remain within the order of thermal fluctuations ($1-4 k_B T$) to obtain unbiased results and be able to extract kinetic and thermodynamic parameters as shown for echinomycin. This might become more challenging for longer k-mers, as hairpins have to accommodate longer AT-rich and GC-rich motifs. The combination of very stable motifs (GCGCGCGC) and very weak motifs (e.g. TTTTTTT) should increase the roughness of the free-energy landscape. A way to obtain unbiased results for longer k-mer motifs is to partition the sequences between two or several hairpins with different levels of GC-content. Otherwise, experiments could also be performed in a single hairpin that contains regions of different thermodynamic stability and using the blocking oligo approach to study the different regions sequentially in a single experiment (by applying an equivalent ΔF test force in each region).

Finally we would like to emphasize that for very long DNA binding motifs, such as those shown for some transcription factors (up to 15 bp), it might be more informative to perform a pre-selection of target binding sites to reduce the size of the combinatorial space being able to study this subset of target binding sequences within different flanking contexts.

We have included these considerations in an extended discussion of the results (page 14 line 381).

[1] Hugué, Josep M., et al. "Single-molecule derivation of salt dependent base-pair free energies in DNA." *Proceedings of the National Academy of Sciences* 107.35 (2010): 15431-15436.

[2] Bockelmann, Ulrich, et al. "Unzipping DNA with optical tweezers: high sequence sensitivity

and force flips." *Biophysical journal* 82.3 (2002): 1537-1553.

6) Minor points:

- Page 3: Insert space between number and unit at "1nm".
- Page 6, top: "we found out a ..." better: "we found a..."
- Page 13: "25 C" better: "25°C"

We thank the reviewer for noting these errors. We have corrected them in the revised version of the manuscript

Peer Reviewer #2:

In this report, Manosas et al. provide a single-molecule approach that uses magnetic tweezers and DNA hairpin unfolding dynamics to characterize the binding sequence, affinity and kinetics of small molecule binding such as DNA intercalators. They apply the same method to describe the DNA binding sequence of large DNA binding proteins such as restriction enzymes. The assay used here is well described and provides high quality data, as the authors have already demonstrated in previous studies. The authors also performed a sound analysis to characterize the DNA binding sequence of the different molecules studied here. Overall, this is a nice biophysical work. However, hairpin opening for molecule binding detection has been used in many previous articles, with either high throughput magnetic tweezers or optical tweezers. In addition to the cited literature, there is also: Shundrovsky et al. *Nat. Struct. Mol. Bio.* 2006, Hall et al. *Nat. Struct. Mol. Bio.* 2009, Jin et al. *Nat. Struct. Mol. Bio.* 2010, Dame et al. *ChemBioChem* 2013, Dittmore et al., *JACS* 2014, Mandal et al. *Angewandte* 2015, Dulin et al. *Biophys. J* 2015. This method is clearly not new in that respect. High-throughput magnetic tweezers (see my comment below) have also been recently used with DNA hairpins to characterize the mechanism of the Tus-Ter interaction (Berghuis et al. *Nat. Chem. Bio.* 2015). Therefore, the methodology used here, though well performed, is not new. Furthermore, this method has not been applied here to answer a biological question of great importance. In conclusion, I think that this work is not appropriate for publication in *Nature Communications*, according to its broad scientific audience. I believe this work is better suited for a more specialized journal, e.g. *Biophysical Journal*.

We thank the reviewer for providing these key references that have helped us correct the previous under-representation of relevant force-spectroscopy work using DNA hairpins to detect binding. We have now included these references in the introduction and discussion sections.

If the editorial board decides to proceed with the paper, the following points need to be addressed.

1) I understand that adding “high-throughput” in the title sounds “hot”, but the authors do not show any evidence for a high-throughput assay: I could not find a graph combining all the acquired traces from a single experiment, or an image of the field of view during a typical experiments, that clearly shows hundreds of tethered beads and few reference beads.

We have added to the supplementary materials and image of the field of view with the beads in a typical experiment.

a) Field of view of a typical experiment. (b) Details of the diffraction image of one bead, at two positions of the magnets, corresponding to a force where the hairpin is closed (low force) and a

force where the hairpin is open (high force). The difference in molecular extension between the two forces can be deduced from the analysis of the diffraction pattern with nm accuracy [1].

[1] Gosse, C. & Croquette, V. *Magnetic tweezers: micromanipulation and force measurement at the molecular level. Biophys. J.* 82, 3314,3329 (2002).

Furthermore, the statistical error represented by the error bars in the histograms and in the bar plots seem relatively large (Fig. 3b, Fig. 5c, Fig. 6a, Fig. 6b), underlying not so large data sets.

In the previous version of the manuscript it was not properly explained how the histograms were computed. For each bead, we first compute the histogram of molecular extension in phase 2, where the blockages are observed (Figure 2a), and convert the extension to base-pairs using the conversion factor (supplementary Figure 4). Next, the histogram is fitted to a number of Gaussian functions (depending on the number of observed peaks) to estimate the blockage positions for each bead (Figure 2b). Finally, we build the distribution of mean blockage positions obtained from many different beads, as shown in Figs. 3, 5 and 6. Those distributions are typically built from the measured blockage positions over 100-200 beads and the error bars are the statistical error that reflects the dispersion between different beads.

In the present version of the manuscript we have improved the discussion on how error bars are determined to clarify the analysis. In particular we have added the following discussion in page 7 line 172 (as well as modifications in the captions of the figures describing how error bars are calculated):

“For each intercalator, we perform the FC protocol for several beads in parallel. For each bead we compute the blockage positions, estimated as the center of the peaks in the histogram of molecular extension during phase 2 (as done for Echinomycin in Fig. 2b). We next build the distribution of blockage positions obtained for all beads (100-200 beads), that show a series of peaks centered at specific DNA locations (Fig. 3a and Supplementary Fig. 5).”

Therefore, if the authors do not provide more evidence, the mention “high-throughput” must be removed from the title. The authors should also have cited publications in the field that have already demonstrated high-throughput force spectroscopy measurements, e.g. de Vlaminck et al. NanoLetter 2011 and PlosOne 2012, Dulin et al. NAR and Cell Rep 2015, Duderstadt et al. Mol Cell 2016... The most recent studies show ~100s to close to 1000 tethered followed simultaneously with a similar spatiotemporal resolution. I believe that the authors should demonstrate such capabilities to maintain the “high-throughput” qualification in their title.

The assay proposed in this paper presents different aspects that contribute to its

high-throughput nature. First, we typically work with 50-100 beads in parallel (as mentioned by the referee similar setups can reach even 1000 beads). Next, the protocol proposed allows testing the same molecule many times, which provides a way of increasing the statistics and the quality of the measurement. Finally, and more important, we propose a way of generating DNA sequences to simultaneously test many binding sequences in the same substrate in an unbiased way using force-spectroscopy. In this line, we show that the use of sequence-optimized hairpins with a flat free-energy landscape allows to quantitatively test all possible tetramers (~256 combinations) in short DNA hairpins (200bp). To the best of our knowledge this kind of sequence optimization has never been carried out before in the single molecule force spectroscopy field. For all this we think that the high-throughput nature of the method is well justified. We have added the references suggested by the reviewer in the introduction of the paper where parallelization of magnetic tweezers experiments is discussed (page 3 line 62). We have also modified the introduction to clarify some of the aspects that we consider justify the high-throughput nature of our approach (page 3 lines 55-62)

2) The study of Echinomycin is well done and well described, with a detailed and clear analysis. Given that the authors are using a hairpin to detect molecule binding, they should cite the following work: Dittmore et al., JACS 2014, Mandal et al. Angewandte 2015, Dulin et al. Biophys. J 2015.

We thank the reviewer for providing these key references that have helped us correct the previous underrepresentation of relevant force-spectroscopy work using DNA hairpins to detect binding. We have included these references in the introduction and briefly discussed the contributions of these works.

3) I could not find any mention of the number of data points per histogram or bar plot represented in this study. The author must provide the statistics for each experiment performed in the study. I recommend them to add a table in the Supplementary Information resuming all the conditions and their respective statistics. A clear description of the methodology for the extraction of the plateau is lacking, as well as how the error bars in the bar plots and the histograms are derived.

We have now included the statistics used in each experiment as well as the description of how the error bars are derived. We have added to the supplementary materials the following figure that describes the method used to analyze the experimental data to obtain the plateaus and the histograms of blockages.

(a) Differential measurement of the molecular extension of bead 1 (with respect the fixed bead) during phase two (F_{test} phase) Z_2 where we subtract the average value of the the molecular extension in phase 3 (F_{high} phase) Z_3 . The results shown are the superposition of the data from 135 cycles. Subtraction of the extension in phase 3 improves the alignment between the data from different cycles. (b) Histogram of the molecular extension in panel (a) that shows different peaks. Conversion from the measured extension to base-pairs leads to the results shown in the Fig. 2b in the main text.

4) Little is mentioned about the restriction enzymes binding conditions and affinity. Given that half of the presented work reports on the detection of bound restriction enzymes, it is necessary to say something about their biochemistry, e.g. to which type belongs the studied restriction enzymes, their respective K_d in the buffer used here, their mechanism of binding and cleaving... I have not been able to find any citation of previous biochemical work on Rsa1 for example. The authors mentioned that Rsa1 is a homodimer that binds symmetrically, head and back being equidistant from the recognition site, and have measured a footprint of ~ 14 bps (l. 200, p.8), but I have found neither citation nor complementary biochemical studies supporting their results or assertion. The same works for the other enzyme. I recommend that the authors include reference(s) that support(s) better their work. If the authors cannot provide supporting literature, they should provide bulk biochemical assays to support their claims.

We agree with the reviewer that a more extensive discussion on the restriction endonucleases used in this work is needed as well as references to previous biochemical characterization work.

Rsa1 is a type II endonuclease that recognizes and cleaves the palindromic sequences GTAC within its recognition site (between the T and A nucleotides). The recognition sequence and cleavage sites of this enzyme have been determined from restriction endonuclease mapping using DNA fragments of known sequence and Sanger sequencing [1]. Although an extensive

biophysical characterization is not available, equivalent results were found for its isoschizomer Afal (isolated from *A. facilis* instead of *R. sphaeroides*) [2]. Both enzymes are known to be type IIP endonucleases, which achieve palindromic recognition by homodimer formation [3,4,5].

Although a crystal structure for RsaI is not available yet, its molecular weight is ~30 kDa (Uniprot Q6SA27) in line with other palindromic restriction endonucleases such as EcoRI and EcoRV. Similarly a protein folding prediction using Phyre 2 shows the formation of a DNA-binding domain at the C-terminus, and a dimerization domain in the N-terminus. These results are in line with other better characterized type IIP endonucleases such as EcoRI and EcoRV, for which crystal structures exist (only <30 restriction endonucleases have been crystallized vs > 2500 isolated type II endonucleases). Finally, the binding of RsaI to DNA as an homodimer was confirmed by the New England Biolab technical service.

On the other hand, both BspCNI (MW: 105 kDa) and MnlI (MW: 38 kDa) are type IIS endonucleases that have a non-palindromic recognition sequence and cleave DNA outside this recognition sequence. For these enzymes, recognition sequence and cleavage site were also determined from restriction endonuclease mapping using DNA fragments of known sequence (lambda-phage, ϕ X174 and M13mp18) and using Sanger sequencing experiments of digested versus undigested fragments [6,7]. Type IIS endonucleases are typically found as monomers in solution [4,5,8]. For MnlI electrophoretic mobility shift assays have determined the affinity constant of binding to DNA to be in the range 5-50 nM [9,10], in agreement with values obtained for other type II restriction endonucleases [11]. We have added this information to the results section and included references on the biochemical characterization of these enzymes.

[1] Lynn, SP, et al. "RsaI: a new sequence-specific endonuclease activity from *Rhodospseudomonas sphaeroides*." *Journal of bacteriology* 142.2 (1980): 380-383.

[2] Dou, D, et al. "Restriction endonuclease Afal from *Acidiphilium facilis*, a new isoschizomer of RsaI: purification and properties." *Biochimica et Biophysica Acta (BBA)-Gene Structure and Expression* 1009.1 (1989): 83-86.

[3] Roberts, R. J., et al. "A nomenclature for restriction enzymes, DNA methyltransferases, homing endonucleases and their genes." *Nucleic acids research* 31.7 (2003): 1805-1812.

[4] Pingoud, A, et al. "Structure and function of type II restriction endonucleases." *Nucleic acids research* 29.18 (2001): 3705-3727.

[5] Pingoud, A., et al. "Type II restriction endonucleases: structure and mechanism." *Cellular and molecular life sciences* 62.6 (2005): 685-707.

[6] Jurenaite-Urbanaviciene, S, et al. "Characterization of BseMII, a new type IV restriction-modification system, which recognizes the pentanucleotide sequence 5'-CTCAG (N) 10/8." *Nucleic acids research* 29.4 (2001): 895-903.

[7] Kamp, D, et al. "Mapping of restriction sites in the attachment site region of bacteriophage lambda." *Molecular and General Genetics MGG* 154.3 (1977): 231-248.

[8] Szybalski, W, et al. "Class-IIS restriction enzymes—a review." *Gene* 100 (1991): 13-26.

[9] Kriukiene, E., et al. "MnII—The member of HNH subtype of Type IIS restriction endonucleases." *Biochimica et Biophysica Acta (BBA)-Proteins and Proteomics* 1751.2 (2005): 194-204.

[10] Kriukiene, E.. "Domain organization and metal ion requirement of the Type IIS restriction endonuclease MnII." *FEBS letters* 580.26 (2006): 6115-6122.

[11] Jen-Jacobson, L. "Protein—DNA recognition complexes: Conservation of structure and binding energy in the transition state." *Biopolymers* 44.2 (1997): 153-180.

I strongly recommend the authors to add in their study a characterization of a well-described restriction enzyme, e.g. EcoRV (also provided by NEB), which will provide them with plenty of literature to compare their results with. This would allow them to validate their mechanical footprinting method.

Although we do not show experiments with EcoRI or EcoRV in this work, we have performed force-spectroscopy experiments with EcoRI restriction enzymes using shorter DNA hairpins (34-bp) in a previous work [1]. In these experiments we also observed DNA blockage events 6-bp ahead of the expected recognition sequence of EcoRI, suggesting that the observed phenomenology is consistent between different type IIP endonucleases. We have added this information and reference to the main text. We have included the following sentence in line 248 page 9:

"This observation is in agreement with results obtained for EcoRI using short (34-bp) DNA hairpins, where blockage events ~6-bp ahead from the recognition sequence are observed, suggesting that a mechanical footprint of approximately half and helical turn might be consistent between different type IIP endonucleases [1]."

[1] Camunas-Soler, J., et al. "Experimental measurement of binding energy, selectivity, and allostery using fluctuation theorems." *Science* 355, 6323 (2017): 412-415.

5) The authors look at the binding affinity of enzymes, but they did not provide any meaningful concentration (only Units/ml). I recommend them to contact NEB, as they purchased all their enzymes from that company, to obtain the concentration of the stocks. This would be meaningful to compare with the Kd on random and specific DNA binding sequence.

We contacted NEB as suggested by the reviewer and obtained the following concentrations for the different enzymes used in this study. Molar concentrations are reported in dimer units for the two enzymes known to bind as dimers to DNA (RsaI and MnII).

Rsal: 50 nM dimer [1.7 µg/ml] (MW: 19 kDa monomer)
BspCN: 2 µM [270 µg/ml] I (MW: 105 kDa)
MnII: 1.2 µM dimer [90 µg/ml] (MW: 38 kDa monomer)

From these values, we derive that the concentrations used for the experiments with MnII are compatible with the K_d values obtained in previous biophysical bulk experiments [1,2]. We have added this information in the methods section (ligands and buffer subsection).

[1] Kriukiene, Edita, et al. "MnII—The member of HNH subtype of Type IIS restriction endonucleases." *Biochimica et Biophysica Acta (BBA)-Proteins and Proteomics* 1751.2 (2005): 194-204.

[2] Kriukiene, Edita. "Domain organization and metal ion requirement of the Type IIS restriction endonuclease MnII." *FEBS letters* 580.26 (2006): 6115-6122.

6) All the enzymatic study performed here use a buffer containing EDTA (TE buffer, see Methods), which traps any residual traces of divalent ions in the solution. I understand that the authors do not want any DNA cleavage during their experiments. However, the binding affinity and specificity of restriction enzymes is strongly dependent on the presence of divalent ions, e.g. EcoRV. Though the presence of Mg will generate DNA cleavage, and is therefore incompatible with the study presented here, the authors can use other divalent ions to conserve the binding affinity observed in the presence of Mg while impairing DNA cleavage. They could look for example at the work published by Bowen and Dupureur, *Biochemistry* 2003 to find such a divalent ion.

As suggested by the reviewer, we have followed his advice and performed experiments with divalent ions. In particular we have chosen to work with CaCl_2 since the work by Bowen et al., show that this divalent ion increases the binding affinity without inducing significant cleavage in type II restriction endonucleases [1].

We have performed experiments at 1 mM CaCl_2 with both BspCNI and MnII. The results are compatible with those obtained in absence of divalent ions: we observe similar distribution of blockages positions and weights of each peaks. The only clear difference observed is in the frequency of blockages. In presence of CaCl_2 , the frequency of blockages increases, revealing the larger binding affinity.

With MnII we also observe that we quickly lose tethered beads after injection of the enzyme, a signature that we are inducing some DNA cleavage. This is in agreement with Ref.[2] where it is shown that Ca^{2+} can induce weak cleavage ($\text{Mg}^{2+} > \text{Ni}^{2+} = \text{Co}^{2+} > \text{Mn}^{2+} > \text{Ca}^{2+} > \text{Zn}^{2+}$). We have included a new Supplementary figure describing these results.

(a,c) Histogram of blockage positions obtained by applying the FC protocol to the H_{S4} hairpin in presence of BspCNI (a, Number of beads=153) and MnlI (c, Number of beads=113) restriction enzyme in presence of 1mM $CaCl_2$. The results show peaks located at the same positions measured in absence of divalent ions (Figure 6 in main text and Supplementary Table 2). Error bars are inversely proportional to the square root of the number of points for each bin. In the case of MnlI, the tethered beads detach shortly after the injection of the enzyme, and indication that calcium ions induce cleavage. (b,d) Frequency of blockages, measured as the average number of blockages per cycle, for BspCNI (b) and MnlI (d) at different $CaCl_2$ concentrations. Error bars correspond to the s.e.m.

[1] Bowen, L. M., and Dupureur, C. M.. "Investigation of restriction enzyme cofactor requirements: a relationship between metal ion properties and sequence specificity." *Biochemistry* 42.43 (2003): 12643-12653.

[2] Kriukiene, E. "Domain organization and metal ion requirement of the Type IIS restriction endonuclease MnlI." *FEBS letters* 580.26 (2006): 6115-6122.

7) On BspCNI, the authors observe two binding sites per binding region (Fig. 6b), and propose that one comes from the recognition site, the other one from the cleavage site. Do they have any biochemical proof to bring such conclusion? Is there anything in the literature about it?

BspCNI is a single-chain R-M-S composite enzyme, meaning that a single restriction enzyme molecule is composed of restriction, methyltransferase, and specificity modules all fused together (MW = 105 kDa). Like other type of IIS restriction endonucleases it cleaves DNA outside its recognition sequence (in this case 10 and 8 nucleotides ahead of it on each strand) [1,2]. The recognition and cleavage sites of its isoschizomer BseMII have been well characterized in previous biochemical assays [3], and the technical service of NEB has also provided alternative confirmation of all this information for BspCNI. Consequently, the fact that BspCNI recognizes the CTCAG motif cleaving DNA 10 bp and 8 bp ahead on each strand seems to be very well established (recognition sequence: CCTC(N)₁₀/GAGTC(N)₈).

In our footprinting experiments we observe two binding events for each target sequence of BspCNI (Fig. 6a, four cyan lines). The first binding event is located 2 bp ahead from the expected cleavage site of the enzyme (first cyan line), whereas the second binding event is located on the recognition sequence of the enzyme (second cyan line).

The second half of the hairpin is the reverse complement of the first half of the hairpin, and therefore the interaction between BspCNI and DNA in the second target sequence is tested in the reverse orientation. Again, in this case we observe a first blockage event 2 bp ahead from the expected recognition sequence of the enzyme (third cyan line) and a second blockage event in the expected cleavage site of the enzyme (fourth cyan line). The relative height of the peaks is compatible for both orientations of the enzyme (i.e. the peak observed closer to the recognition site is higher than the peak observed closer to the cleavage site). We do not observe any binding events >20 bp away from the expected recognition sites and cleavage sites of the enzyme, that could be explained by random non-specific binding of the enzyme to other DNA sequences.

From our point of view the simpler explanation for this phenomenology is that the interaction between BspCNI and DNA in the recognition site and cleavage site are being tested when the hairpin/DNA complex is mechanically disrupted. These results are also compatible with similar force-spectroscopy experiments where the asymmetric binding of heterodimeric DNA mismatch repair complexes is tested using independent forward and reverse hairpins [3]. Nevertheless, and as shown in our answer to the next question below, we performed experiments at a different enzyme concentration to provide further evidence that the observed phenomenology is not related to non-specific binding to different sequences.

[1] Szybalski, W., et al. "Class-IIS restriction enzymes—a review." *Gene* 100 (1991): 13-26.

[2] Jurenaite-Urbanaviciene, S., et al. "Characterization of BseMII, a new type IV restriction–modification system, which recognizes the pentanucleotide sequence 5'-CTCAG (N) 10/8." *Nucleic acids research* 29.4 (2001): 895-903.

[3] Jiang, J., et al. "Detection of high-affinity and sliding clamp modes for MSH2-MSH6 by single-molecule unzipping force analysis." *Molecular cell* 20.5 (2005): 771-781.

Did they probe whether this could originate from non-specific DNA binding of the enzyme? I believe that the authors should provide experiments with at least another concentration of BspCNI to demonstrate that it is not non-specific binding. Adding a divalent ion that promotes binding but impair cleavage would help to show that it is the cleavage site that binds.

We have performed experiments at a lower concentration and we obtained similar results: two peaks are observed for a single recognition site, close to the restriction and recognitions sites (see Figure below). On the other hand, the frequency of blocking decreases strongly (by a factor 4) when decreasing the enzyme concentration (by a factor 5), showing that we are not at saturating conditions.

(a) Histogram of blockage positions obtained by applying the FC protocol to the H_{s4} hairpin in presence of 4 nM of BspCNI restriction enzyme (Number of beads=99). The results show peaks located at the same positions measured in absence of divalent ions (Figure 6 in main text and Supplementary Table 2). Error bars are inversely proportional to the square root of the number of points for each bin. (b) Frequency of blockages, measured as the average number of blockages per cycle at two different BspCNI concentrations. Error bars correspond to the s.e.m.

8) To definitely demonstrate the mechanical footprint of BspCNI, I recommend the authors to perform an experiment similar to the one described in Fig. 6a, but with the second restriction site inverted, to determine whether the distance between the two Gaussian peaks in Fig. S8c varies. The first two peaks (Fig. 6a, ~150 and 160 bp) representing the binding on the first site should remain unchanged, providing an absolute calibration in position, when the second should demonstrate the asymmetry of the binding of BspCNI. The same works for MnlI. This will further demonstrate the quality of their calibration.

In our experiments with restriction endonucleases we had already introduced a second

restriction site inverted in the hairpin sequence, therefore implementing the suggestion by the reviewer. In fact, our originally designed DNA hairpin construct contains a forward sequence with all possible 4-mers, followed by its reverse sequence (hairpin S_4S_4). In this way, every protein-DNA interaction can be mechanically disrupted from both orientations in a single experiment. In the revised manuscript we have now explicitly indicated that the second half of the hairpin is the reverse sequence of the first half of the hairpin (see the new hairpin schematics in Figures 6a and 6b). We hope that this fact is now clearer to the reader. For BspCNI only one binding site is present in the original forward sequence and therefore the interaction is only at one location for each orientation ($==<==$ | $==>==$). On the other hand, the recognition sequence of MnlI is present twice in the original forward sequence, and therefore the interaction is found two times for each orientation ($==>==<==$ | $==>==<==$). We think that such experiment provides the required configurations to demonstrate the asymmetry of binding of an enzyme, in a similar way as has been performed for DNA repair mismatch complexes using two independent hairpins [1]. Note that in order to perform experiments with such a forward-reverse construct we have introduced a spacer sequence that avoids misfolding into a cruciform structure.

[1] Jiang, J., et al. "Detection of high-affinity and sliding clamp modes for MSH2-MSH6 by single-molecule unzipping force analysis." *Molecular cell* 20.5 (2005): 771-781.

9) In the last paragraph of the Results section (l.239-246), the authors rightly highlight a difference in the weight of each Gaussian from Fig. 6b and Fig. S10c using MnlI. They explain that they can discriminate the origin of this difference from the similar binding region and flanking sequences that are inverted in the hairpin S4. As they use the same sequence, I would conclude that the origin of the difference in weight comes from the protein orientation. However, they conclude that the difference comes from the flanking sequence. This needs to be rewritten because it is unclear what the authors conclude here.

We thank the reviewer for the suggestion. We have rewritten the results section describing this experiment to make clearer that the observed asymmetry on the peak weight for MnlI is caused by the different flanking sequence around each recognition site, instead of being driven by the pulling orientation of the experiment (i.e. from which end the protein/DNA complex is mechanically disrupted). The new description is found in page 11, line 309.

10) The authors should provide a statistical analysis of the restriction enzyme blocking time, as they have performed for the Echinomycin. This will inform the reader about the mechanism of release of the enzyme on the DNA sequence upon DNA unzipping.

The lifetime blockage experiments done for Echinomycin represent a proof of principle of additional measurements that can be obtained in the single-molecule footprinting approach that we developed. These experiments are complementary to the main scope of the paper, that is

the determination of the preferred binding sequences on an unbiased way in force-spectroscopy experiments for small ligands and/or proteins. Consequently the characterization of Echinomycin is more complete (e.g. different hairpin sequences, testing forces, equilibrium experiments) than the analysis done for the other ligands.

On the other hand, we think that the experiments with non-palindromic restriction enzymes are particularly useful to illustrate that the presented method allows to detect asymmetries between the recognition sequences of an enzyme and its mechanical footprint, an effect that is unobservable both for small ligands and for usual palindromic endonucleases (e.g. EcoRI, EcoRV). Such a measurement would be also difficult to obtain using bulk rather than force-spectroscopy based methods. This is the main reason why we focused our study on experiments with non-palindromic endonucleases.

Although it would be also interesting to perform blockage lifetime experiments with the restriction endonucleases studied here as the referee suggests, this would require to repeat all the experiments for each restriction enzyme at different forces, a task which is beyond the scope of the paper.

11) The description of the correlation analysis to determine the recognition sequence in the Methods section is concisely and well explained but hard to understand without drawing it. I think that it is necessary for the authors to include a schematic describing this analysis in the Supplementary Information.

As suggested by the reviewer, we have added new figure panels in the supplementary describing the procedure:

(a) Schematics of the computation of the similarity function defined in Methods in the main text,

that is used to perform a sequence correlation analysis. (b) Results for the similarity function S_{ij} defined in Methods. The scale of blue denotes the degree of sequence similarity: the darkest blue color accounts for maximum similarity (e.g. all bases coincide).

12) The results show here the binding DNA sequence for well-known restriction enzymes (they are referenced on Wikipedia). How does this method perform with enzymes that have an unknown binding sequence? It is necessary to add such a study to fully demonstrate the generality of this method.

Although all the enzymes used in this work are commercially available from New England Biolabs, most of the biophysical information available for these enzymes is scarce and restricts to its recognition sequence and cleavage sites as obtained from mapping digest and Sanger sequencing (as also noted by the reviewer in previous questions) [1,2,3]. Our work is the first one to directly show the coverage size of these enzymes on DNA through its mechanical footprint. We have done so by studying both palindromic type II restriction endonucleases and more unusual type IIS endonucleases, where asymmetric binding can be observed. Furthermore similar work on a more well characterized enzyme (EcoRI) has been reported elsewhere by ourselves [4], proving that our method works well not only for small DNA ligands but also for large proteins. We agree that characterizing enzymes with unknown binding sequences is of much interest but we are not convinced that this would necessarily strengthen the evidences already provided in the manuscript in its present form.

[1] Lynn, Steven P., et al. "RsaI: a new sequence-specific endonuclease activity from *Rhodospseudomonas sphaeroides*." *Journal of bacteriology* 142.2 (1980): 380-383.

[2] Jurenaite-Urbanaviciene, S., et al. "Characterization of BseMII, a new type IV restriction–modification system, which recognizes the pentanucleotide sequence 5'-CTCAG (N) 10/8." *Nucleic acids research* 29.4 (2001): 895-903.

[3] Kamp, D., et al. "Mapping of restriction sites in the attachment site region of bacteriophage lambda." *Molecular and General Genetics MGG* 154.3 (1977): 231-248.

[4] Camunas-Soler, J., et al. "Experimental measurement of binding energy, selectivity, and allostery using fluctuation theorems." *Science* 355.6323 (2017): 412-415.

Peer Reviewer #3:

In this manuscript, Manosas et al. describe a method for mapping bound ligands by unzipping DNA using magnetic tweezers that can simultaneously monitor multiple tethered beads. Although DNA unzipping for mapping methods has been previously demonstrated, this manuscript distinguishes itself by systematically designing DNA sequences for constant force experiments and providing detailed studies of several

classes of ligands and binding proteins. I would like to recommend publication in Nature Communications if the authors are able to fully address the following comments.

1) It seems that the authors might not be aware of a relevant publication for DNA foot-printing using unzipping DNA (Jiang et al., Detection of high-affinity and sliding clamp modes for MSH2-MSH6 by single-molecule unzipping force analysis. Mol Cell, 2005, 20:771-81). Jiang et al. mapped the footprint of the DNA repair enzyme by unzipping a single DNA molecule from both the forward and reverse directions, similar to the current manuscript (figures 5 and 6). Please include a discussion of comparison with this work.

We thank the reviewer for providing this reference that we were not aware of. We have added the following sentence in page 10, line page 265 (where the forward-reverse hairpin strategy is introduced).

“A similar approach using independent forward and reverse hairpin constructs has been successfully used to prove asymmetric binding of protein heterodimers in DNA mismatch repair [1].”

[1] Jiang, Jingjing, et al. "Detection of high-affinity and sliding clamp modes for MSH2-MSH6 by single-molecule unzipping force analysis." *Molecular cell* 20.5 (2005): 771-781.

Finally, we have also added the reference to the discussion of the results, where we introduce previous work on systems where asymmetric binding is relevant.

2) There is another relevant publication that also seems to have been overlooked (Koch et al., Dynamic force spectroscopy of protein-DNA interactions by unzipping DNA double helix. Phys. Rev. Lett. 91:028103, 2003). Koch et al. determined the equilibrium binding constant of a bound protein K_d , the rate of dissociation k_{off} , and the distance of the bound state to the activation barrier by using DNA unzipping dynamic force spectroscopy. Again, please a discussion of comparison with this work.

We agree that a comparison to previous quantitative experiments using dynamic force spectroscopy should be added to the discussion of the results. We have included a sentence in the results section (page 6, line 151) and in the final discussion of the results respectively (page 12, line 342). We have also included the above reference at several places in the introduction and discussion of the results.

New sentence added to the results section:

“These values for X^\ddagger for echinomycin are significantly larger than those previously obtained for restriction endonucleases in dynamic force spectroscopy experiments ($X^\ddagger \sim 1$ nm) [1], showing that small ligand interactions are more sensitive to force-induced unbinding than large protein-DNA complexes.”

New sentence added to the discussion section:

“We also develop protocols to extract binding energies, the kinetic unbinding rate and the position of the transition state (X^\ddagger), similarly as previously shown for protein-DNA interactions using dynamic force spectroscopy [1].”

[1] Koch, Steven J., and Michelle D. Wang. "Dynamic force spectroscopy of protein-DNA interactions by unzipping DNA." *Physical Review Letters* 91.2 (2003): 028103.

3) The authors stated that the magnetic tweezers instrument provides a resolution of ~ 2 bp. However, data in figure 2a suggest significant drift over time, and figure 2b also shows peak widths that are much broader than 2 bp. Please show evidence (which can be included in the SI) for how this 2 bp resolution is determined. Related to this, the DNA attachment point on a magnetic bead varies from bead to bead. How is the absolute extension determined? Please include a discussion under Methods.

The drift over time is compensated doing differential measurements. First, we use the extension of a fixed bead (attached non-specifically to the glass surface) to perform differential measurements and reduce the drift. Second, at each cycle we subtract the extension in phase 3 (F_{high} phase) to that of phase 2 (F_{test} phase), in order to achieve a better alignment between different cycles (see new supplementary figure 3). The histogram of molecular extension in phase 2 in Figure 2b is then fitted to Gaussian functions that have a width of 2 bp (New supplementary table 2). The reviewer should keep in mind that a root mean square fluctuation of 2 bp means that on some occasion you will see fluctuation of 4 to 6 bases. As suggested by the referee a discussion on the extension measurements has been added to the methods section.

4) The theory of dynamic force spectroscopy was used to find k_0 , which was obtained as the rate at zero force of unzipping. However, the unzipping force itself is ~ 15 pN, below which force no longer has any impact on the lifetime of a bound ligand. Therefore, it would seem that k_0 should not be obtained at force = 0, and instead at force = 15 pN. What is the reason to extrapolate the force to zero?

As the referee highlights, at forces below 15pN it is not possible to observe force-induced unbinding events, as the folded hairpin configuration is most stable and the hairpin does not spontaneously unfold at these forces. This does not allow us to measure the lifetime of the

bound DNA complexes at forces lower than the average unzipping force. However, the effect of an applied force on the ligand-DNA complex should still follow the Kramer's Bell-Evans theory in a similar way as shown in reference [1] for DNA binding proteins. Consequently if we had an alternative way to measure unbinding events (e.g. fluorescence) while applying forces lower than the average unzipping force of the hairpin (<15 pN) to the DNA-ligand complex, the lifetime of the bound complexes should still follow Bell-Evans. In other words, although a force below 15 pN is not enough to unfold the hairpin in experimental timescales, we expect the force-dependence of the ligand-DNA interaction to be still described by a thermally-activated process where the activation barrier is lowered by the application of force.

[1] Koch, Steven J., and Michelle D. Wang. "Dynamic force spectroscopy of protein-DNA interactions by unzipping DNA." *Physical review letters* 91.2 (2003): 028103.

5) The authors wrote in lines 133-134: "Interestingly, the unbinding rate associated to the last binding site corresponds to the product of unbinding rates for single ligands, suggesting that the binding of two contiguous Echinomycin bis-intercalators is cooperative." This does not seem to make sense based on unit analysis. The unbinding rate has units of 1/s, and the thus the product of two rates would have units of 1/s². These two quantities cannot be the same. Please clarify.

We thank the reviewer for raising the attention of this subtle point. In the absence of cooperativity we expect the total binding energy of two consecutive ligands to be equal to the sum of their individual binding energies ($\Delta G_{AB} = \Delta G_A + \Delta G_B$). In the presence of cooperativity an additional energetic contribution $\Delta\Delta G_{AB}$ should be added. The affinity constant (K) of each of the interactions is defined as $K/C_0 \sim \exp(-\Delta G^0/kT) = k_{on}/k_{off}$, where C_0 is the ligand concentration and k_{on} and k_{off} are the kinetic on and off-rates respectively (in units of [1/s]). If one assumes that the kinetic on-rate (k_{on}) does not change in the presence of a neighbouring ligand, one would then expect that in the absence of a cooperative term (i.e. when $\Delta G_{AB} = \Delta G_A + \Delta G_B$), the following relation holds: $k_{off,AB} = k_{off,A} k_{off,B} / k_{on}$, as derived from $\exp[(-\Delta G_A - \Delta G_B)/kT] = K_A K_B / (C_0^2) \sim (k_{on}^2) / (k_{off,A} * k_{off,B})$. In order to unambiguously prove that this relation does not hold for echinomycin, and that an additional $\Delta\Delta G_{AB}$ term is needed to explain the measured kinetic on-rates and off-rates, we would need to measure the kinetic on-rates of the reaction (which we did not measure in the current set of experiments). Although we cannot conclude cooperative binding from the measured kinetic off-rates, we have also measured a change in the position of the transition state (X^\ddagger) for the two contiguous binding sites ($X^\ddagger \sim 4$ -bp) in relation to individual binding sites ($X^\ddagger \sim 2$ -bp) (Figure 2d,e). This increased distance of the transition state, suggests that the two ligands are unbound on a single cooperative step. Moreover a cooperative term between contiguous echinomycin ligands has been previously observed in biochemical experiments performed in equilibrium conditions [1]. This is supported by our own recent studies that show a similar result using non-equilibrium methods [2]. Still, the kinetic off-rates results measurements shown in the current manuscript do not allow us to unambiguously support this statement and we have modified the text accordingly. We thank the reviewer for calling our attention to this point.

On the other hand, from a dimension analysis point of view, when saying that the total unbinding rate is the product of two intermediate unbinding rates, what we actually mean is that the overall probability of unbinding of the ligand is the product of the probabilities of two consecutive unbinding events, and probabilities are dimensionless.

[1] Bailly, C., François H., and Waring, M. J.. "Cooperativity in the binding of echinomycin to DNA fragments containing closely spaced CpG sites." *Biochemistry* 35.4 (1996): 1150-1161.

[2] Camunas-Soler, J., et al. "Experimental measurement of binding energy, selectivity, and allostery using fluctuation theorems." *Science* 355.6323 (2017): 412-415.

6) For Figure 4c, the detected peaks are not located at the same locations relative to the XCGY sites. What is the cause for this misalignment?

The referee raises a very interesting point that was not properly discussed in the previous version of the manuscript. In the sequence selectivity experiments with echinomycin (Fig 4C) the location of the blockage events (cyan lines) is shifted 2-4 bp earlier to the expected CG binding motif at some binding locations. This fact correlates with binding peaks that have a larger width (i.e. more spread). This is particularly apparent in peaks 3 and 4 of Fig 4c. We interpret this as a signature of non-specific binding to secondary binding sites located nearby to a primary binding site (CG dinucleotide motif, also called CG-step). It is known that echinomycin binds non-specifically to DNA, having preference for GC-rich sequences [1,2,3]. This might be particularly important in regions containing a GC-rich stretch ahead of a strong CG-step (e.g. 5'-CCTTCGA-3') where two echinomycin ligands might be bound one next to each other (i.e. to the dinucleotides CC and to CG), as observed in previous bulk footprinting experiments [1]. For some unzipping curves, a ligand might be bound to the secondary binding site ahead of the CG-step giving a blockage event ahead of the expected CG-step. For other pulling curves only the expected blockage at the high affinity CG-step is observed. This should give broader peak distributions at these binding regions vs sharp peaks in regions where the CG-step is embedded in AT-rich regions (which show little nonspecific binding for echinomycin) [1,2]. This interpretation is in agreement with a similar effect observed in Figure 3a for the peak containing two consecutive high affinity binding sites (peak 3, TCGTACGA), where a broader Gaussian distribution is also observed (width ~4 nm for the double binding site vs width ~2 nm for the single binding sites).

Two other potential interpretations that could contribute to shifts on the location of blockage events are (1) experimental errors in the measurement and (2) local free-energy landscape effects due to the presence of a GC-rich sequence stretch ahead of a binding site. In other words, the presence of a GC-rich stretch ahead of a binding site might shift the position of the barrier to unfolding a few basepairs ahead of the expected binding site (i.e where the initial GC-rich stretch starts). Although we think that these effects are quite unlikely (we have designed the DNA hairpins to have barriers not larger than $3 k_B T$) we cannot completely rule out this interpretation. Nevertheless, the fact that deviations from the expected binding sites are mostly apparent at locations where we expect multiple binding of ligands, suggests that these

two additional sources of error are within the resolution of the mechanical footprinting experiments (~2 bp).

We have modified the results and discussion section accordingly to explain this effect.

[1] Van Dyke, Michael M., and Peter B. Dervan. "Echinomycin binding sites on DNA." *Science* 225 (1984): 1122-1128.

[2] Leng, Fenfei, Jonathan B. Chaires, and Michael J. Waring. "Energetics of echinomycin binding to DNA." *Nucleic acids research* 31.21 (2003): 6191-6197.

[3] Camunas-Soler, J., et al. "Experimental measurement of binding energy, selectivity, and allostery using fluctuation theorems." *Science* 355.6323 (2017): 412-415.

REVIEWERS' COMMENTS:

Reviewer #1 (Remarks to the Author):

The authors have addressed all my outstanding comments during the revisions and have improved the manuscript by the changes made.

I recommend the manuscript now for publication.

Reviewer #2 (Remarks to the Author):

The authors of "Single molecule high-throughput footprinting of small and large DNA ligands" have provided a very complete rebuttal. I particularly appreciate the effort they made to provide new experiments that support their claims and to dig into the biochemical literature of restriction enzymes. Therefore, I am entirely satisfied with the new version of the manuscript and I fully support the publication of the article into Nature Communications.

Reviewer #3 (Remarks to the Author):

I am satisfied with the response from the authors and now recommend publication in Nature Comm.